# ALIENLM: ALIENIZATION OF LANGUAGE FOR PRIVACY-PRESERVING API INTERACTION WITH LLMS

## ABSTRACT

We introduce **AlienLM**, a framework that reinterprets encryption as language translation for large language models accessed exclusively through black-box APIs. Existing approaches based on secure inference or differential privacy and federated learning offer limited protection in API-only scenarios. **AlienLM** constructs an Alien Language through a vocabulary-level bijection and employs API-only fine-tuning, thereby ensuring compatibility with commercial black-box services while requiring no access to model internals. Across four LLMs and seven benchmarks, **AlienLM** preserves more than 81% of the original performance, and exhibits strong robustness against token-mapping and frequency-analysis attacks. **AlienLM** provides a deployable, low-overhead mechanism for safeguarding sensitive data in API-mediated applications such as healthcare, finance, and education. More broadly, our findings reveal a practical separation between linguistic representation and task competence, thereby motivating future work on composable privacy-preserving layers and formal characterizations of the learnability–opacity trade-off.

## 1 INTRODUCTION

Large language models (LLMs) are now widely deployed across industries and research domains, raising pressing concerns about protecting sensitive information. In particular, global regulatory regimes, such as GDPR in the EU, NIST guidance in the US, and PIPA, APPI, PDPA, and the DPDP Act in Asia, increasingly emphasize encryption as a primary safeguard. This motivates the need for a practical encryption framework that can preserve the confidentiality of prompts, outputs, and training data even when using API-based external LLMs. A more detailed overview of regional regulatory requirements is provided in Appendix A.1.

Privacy-preserving approaches largely fall into two families: (i) *secure inference* based on cryptography and secure computation such as fully homomorphic encryption (HE), garbled circuits (GC), secure multi-party computation (MPC), and trusted execution environments (TEEs) (Gilad-Bachrach et al., 2016; Juvekar et al., 2018; Mishra et al., 2020) and (ii) privacy-preserving training such as differential privacy (DP) and federated learning (FL) (Abadi et al., 2016; Li et al., 2022; Yao et al., 2024). The former often incurs latency and communication overhead and assumes access to model internals or specialized runtimes, which clashes with commercial black-box API settings. The latter primarily targets training data and offers limited confidentiality for prompts and outputs at inference time. In short, under weight-private, black-box API constraints, practical methods that operate purely at the text level while balancing security and utility remain scarce.

These limitations are particularly acute when the provider withholds model internals in API-based services. Applying prior methods either exposes model details externally or fails to protect one of training or inference data, creating a dilemma between data owners who seek to protect sensitive inputs and service providers who avoid revealing parameters. As Knodel et al. (2024) note, combining end-to-end encryption (E2EE) with AI models can introduce security and legal compatibility frictions.

To close this gap, we propose **AlienLM**. The key idea is to reformulate encryption as language translation. Using only the publicly available information that is tokenizer and vocabulary, we construct

an *Alien Language* by applying a bijective permutation to the base vocabulary and adapt the model to this new language via API-only fine-tuning, which we call Encryption Adaptation Training (EAT). Concretely, we (i) define *alienization* to minimize human readability while preserving LLM learnability, and (ii) introduce an alien language construction algorithm that jointly optimizes embedding similarity and edit-distance–based opacity.

With Alien Language, we can layer text-level encryption on top of off-the-shelf tokenizers or vocabularies and apply it to a range of API-based LLMs without accessing internal weights. Our contributions are:

- **Bijection-based encrypted language layer:** We define an *Alien Language* and a *translator* built from a vocabulary-level bijection, enabling bidirectional and lossless conversion between plaintext and alien text (ciphertext) while keeping model internals hidden.
- **API-only adaptation (EAT):** Through API-only fine-tuning, the model adapts to the new language and consistently preserves over 80% of original performance, yielding the Alien Language adapted model, $\mathcal{M}_{\text{alien}}$.
- **Domain adaptation:** Domain-specific EAT further improves target-task performance. We analyze the balance with general capabilities in code and math domains.

## 2 RELATED WORKS

**Privacy-preserving inference/training via cryptography and secure computation**
For API-based LLM usage, approaches fall into (i) cryptographic/secure-computation secure inference and (ii) privacy-preserving training at the data and pipeline level. The former combines HE and GC (2PC/GC) to protect the model and input pair. Gilad-Bachrach et al. (2016) demonstrated inference over HE, Juvekar et al. (2018) reduced latency via an HE+GC hybrid and Mishra et al. (2020) proposed a system with practical compute overheads. TEE-based solutions are also active but retain trust and performance assumptions. More broadly, combining E2EE with AI may conflict with required security and legal properties (Knodel et al., 2024).

The latter family focuses on training data. DP fine-tuning atop large pretrained models has been explored (Li et al., 2021), and guidance for privacy in labeling has been proposed (Yu et al., 2024). FL is increasingly used to combine siloed datasets (Yao et al., 2024; Ye et al., 2024), but it does not hide prompts and outputs from third parties or the server at inference time. In summary, these methods typically assume white-box access or specialized runtimes such as HE, GC, and TEE. Also, they are limited to the training phase. They do not directly address inference-time confidentiality in a black-box API setting. *AlienLM* fills this gap by providing text-level encryption using only publicly available information , tokenizers and vocabularies.

**Obfuscation/substitution-based protections and their limits**
Another line of work lowers human interpretability while retaining model usability via transformations of text/code. In code, DOBF were proposed pretraining objectives targeting deobfuscation (Roziere et al., 2021), and CodeCipher perturbs the embedding matrix to learn token-confusion maps (Lin et al., 2024).

Closest to our setting, Mishra et al. (2024a) fine-tunes models to handle encrypted inputs, but requires modifying internal embedding and LM head layers, making direct application to black-box APIs difficult. Lin et al. (2025) uses emojis for API-side encryption, but the expressivity limits the range of feasible tasks. Conversely, *AlienLM* extends the combinatorial scope by utilizing subword-level transformations over $10^5$ scale bijections and employs EAT to adjust models to the new language, yielding robustness against traditional frequency and $n$-gram attacks.

**Language/tokenizer adaptation and representation alignment**
Recent evidence suggests that language ability and task competence can be separable within LLMs (Chen et al., 2023; Deng et al., 2025; Huben et al., 2024). Methods that learn new languages while preserving task skills have therefore been explored. Byte or character-level models reduce tokenizer dependence and are robust to noise and multilingual input (Xue et al., 2022; Clark et al., 2022; Tay et al., 2022), and vocabulary transfer or replacement techniques have been proposed (Remy et al., 2024; Minixhofer et al., 2024). On the alignment front, representational similarity across models

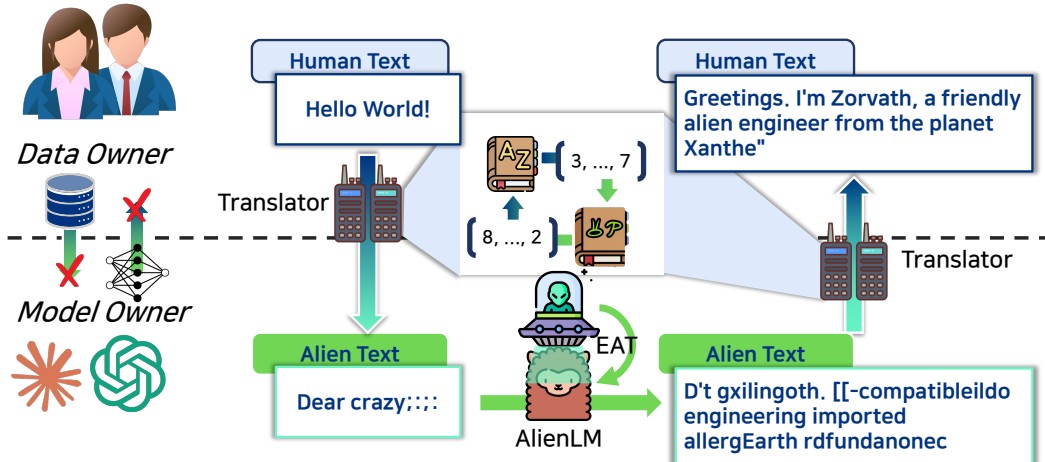

Figure 1: Overview of **AlienLM**. Human text is translated into Alien Text via a vocabulary-level bijection and processed by the API model. The output is then translated back to human-readable text. Note that the same token IDs are shared across human and alien vocabularies, but due to the permuted mapping they decode to different strings.

has been studied (Kornblith et al., 2019), and model stitching quantifies cross-network compatibility (Bansal et al., 2021).

Building on these insights, our work constructs a new formal language, Alien Language, using only a vocabulary bijection and a translator, without white-box access, and acquires it via API-only fine-tuning. Consequently, **AlienLM** attains balance between privacy and utility using only API-exposed components while maintaining the model's original task competence.

## 3 METHOD

**Overview: encryption as language translation.** We reinterpret encryption as a formal-language translation. Using only the public tokenizer $\tau_{\text{target}}$ and vocabulary, we construct an **Alien Language** via a vocabulary bijection $f : I \rightarrow I$, where $I$ is the set of non-special token IDs. This enables lossless client-side encryption and decryption:

$$E_\rho(x) = \tau_{\text{target}}^{-1}\big(f_\rho(\tau_{\text{target}}(x))\big), \quad D_\rho(x') = \tau_{\text{target}}^{-1}\big(f_\rho^{-1}(\tau_{\text{target}}(x'))\big),$$

where $\rho \in [0, 1]$ controls the encryption ratio and $D_\rho(E_\rho(x)) = x$. The server processes alien text using the original tokenizer, while authorized clients translate between plaintext and alien text.

### 3.1 ALIEN LANGUAGE CONSTRUCTION

**Formal language via vocabulary bijection.** For encryption in API-based LLMs, an ideal translation scheme must satisfy three criteria: (i) *API usability* that operates solely with public tokenizers without white-box access, (ii) *human opacity* that minimizes readability to humans, and (iii) *LLM learnability* that preserves semantic relationships from the model's perspective. We achieve these criteria by constructing an Alien Language, which is a formal language in the computer-science sense: a set of strings over a finite alphabet with grammatical rules (Chomsky, 1956; Hopcroft et al., 2001).

Since building a formal language from scratch requires both comprehensive vocabulary coverage and a complete grammatical system, we instead instantiate the Alien Language via a token-ID bijection over an existing vocabulary. This approach inherits the original grammar and expressiveness while altering only the surface form, naturally satisfying all three criteria. It requires only public vocabularies (API usability), produces unreadable token sequences (human opacity), and maintains the underlying semantic structure for model adaptation (LLM learnability).

**Vocabulary and bijection.** Let $v$ be a token string and $i$ its ID. Denote the target model's vocabulary by $\mathcal{V}_{\text{target}} = \{(v_k, i_k)\}_{k=1}^{|\mathcal{V}_{\text{target}}|}$. Let $\mathcal{S} \subseteq \{v_k\}$ be the set of special tokens that must not be replaced, and define $I = \{\, i_k \mid (v_k, i_k) \in \mathcal{V}_{\text{target}}, \ v_k \notin \mathcal{S} \,\}$. We introduce a bijection $f : I \to I$, and define the *alien vocabulary*

$$\mathcal{V}_{\text{alien}} = \{(v_k, \tilde{i}_k)\}_{k=1}^{|\mathcal{V}_{\text{target}}|}, \qquad \tilde{i}_k = \begin{cases} f(i_k), & v_k \notin \mathcal{S}, \\ i_k, & v_k \in \mathcal{S}. \end{cases}$$

**Tokenizer compatibility.** Let $\tau(x; \mathcal{V})$ map text $x$ to token IDs $i$, and $\tau^{-1}(i; \mathcal{V})$ map IDs to text using vocabulary $\mathcal{V}$. For the original tokenizer $\tau_{\text{target}}(\cdot; \mathcal{V}_{\text{target}})$ and the alien tokenizer is defined as $\tau_{\text{alien}}(x; \mathcal{V}_{\text{alien}}) = f(\tau_{\text{target}}(x; \mathcal{V}_{\text{target}}))$.

We define client-side translation over the target tokenizer:

$$E_\rho(x) = \tau_{\text{tgt}}^{-1}\big(f_\rho(\tau_{\text{tgt}}(x))\big), \quad D_\rho(x') = \tau_{\text{tgt}}^{-1}\big(f_\rho^{-1}(\tau_{\text{tgt}}(x'))\big),$$

so that $D_\rho(E_\rho(x)) = x$.[1]

## 3.2 TRANSLATOR: CLIENT-SIDE ENCRYPT/DECRYPT WITH TEXT ONLY

Let $I$ exclude special token set $\mathcal{S}$. Given $\rho \in [0, 1]$, choose $I_\rho \subseteq I$ with $|I_\rho| = \lfloor \rho|I| \rfloor$ and define

$$f_\rho(i) = \begin{cases} f(i), & i \in I_\rho, \\ i, & i \notin I_\rho, \end{cases} \quad E_\rho(x) = \tau_{\text{target}}^{-1}\big(f_\rho(\tau_{\text{target}}(x))\big), \quad D_\rho(x') = \tau_{\text{target}}^{-1}\big(f_\rho^{-1}(\tau_{\text{target}}(x'))\big).$$

Then $D_\rho(E_\rho(x)) = x$. Increasing $\rho$ improves human opacity but may degrade performance.

## 3.3 OBJECTIVE FOR THE BIJECTION: TARGET-EMBEDDING CONSTRAINED DESIGN

**Problem setup.** The bijection $f$ should satisfy the criteria of the ideal translation scheme in Sec. 3.1, unreadable to humans yet learnable by the model. Let $s(i)$ denote the string for token ID $i$, and define the normalized edit distance $\tilde{d}_{\text{edit}}(a, b) = \frac{d_{\text{edit}}(a, b)}{\max\{|a|, |b|\}}$. Let $e_{\text{tgt}}(\cdot)$ be the target model's embeddings and $d_{\text{sim}}$ a similarity-based distance. Over an active domain $I_\rho$, we formulate this as:

$$\max_{f \in \mathfrak{S}(I_\rho)} \quad \sum_{i \in I_\rho} \tilde{d}_{\text{edit}}\big(s(i), s(f(i))\big) \tag{1}$$

$$\text{s.t.} \quad d_{\text{sim}}\big(e_{\text{tgt}}(i), e_{\text{tgt}}(f(i))\big) \leq \alpha, \ \forall i \in I_\rho,$$
$$f(i) \neq i, \ \forall i \in I_\rho, \qquad f(j) = j, \ \forall j \in \mathcal{S} \cup (I \setminus I_\rho).$$

**Lagrangian relaxation.** Relaxing the similarity constraint in equation 1 with multiplier $\lambda \geq 0$ yields the equivalent objective

$$\max_{f \in \mathfrak{S}(I_\rho)} \sum_{i \in I_\rho} \tilde{d}_{\text{edit}}\big(s(i), s(f(i))\big) - \mu \cdot d_{\text{sim}}\big(e_{\text{tgt}}(i), e_{\text{tgt}}(f(i))\big), \qquad \mu = \frac{\lambda}{|I_\rho|}, \tag{2}$$

so larger $\mu$ prioritizes LLM learnability while smaller $\mu$ favors human opacity.

**Proxy embedding.** In a black-box API setting we cannot access $e_{\text{tgt}}$. We therefore approximate it with embeddings from an open-source LLM, $e_P$, replacing $e_{\text{tgt}}$ by $e_P$ in equation 2. This approximation leverages observed cross-model representation alignment (Kornblith et al., 2019; Bansal et al., 2021; Remy et al., 2024; Minixhofer et al., 2024), where relative similarities between tokens are largely preserved across models despite different absolute embedding values. Since the target and proxy models may use different vocabularies, we decompose a target token $v$ into proxy subpieces $S(v) = \tau_{\text{proxy}}(v, \mathcal{V})$ and average: $e_P(v) = \frac{1}{|S(v)|} \sum_{u \in S(v)} e_P(u)$.

---

[1]In practice, our translator composes both the original tokenizer and an alien tokenizer induced by the permuted vocabulary, more details are provided in Appendix A.8.

### 3.4 Approximate Solver for $f$

Solving the bijection exactly is impractical at current LLMs vocabulary scale ($|I_\rho| \approx 10^5$). We therefore use greedy search based on $k$-nearest neighbors (k-NN) candidate reduction.

**Pair score.** We define the pairwise score corresponding to equation 2 as

$$S(i, j) = \tilde{d}_{\text{edit}}(s(i), s(j)) - \mu\, d_{\text{sim}}(e_\star(i), e_\star(j)),$$

where $e_\star$ is ideally $e_{\text{tgt}}$ but practically $e_P$.

**k-NN candidate reduction and greedy pairing.** Direct optimization of Eq. 2 is dominated by the similarity term. While most vocabulary tokens differ substantially in surface form, they may share similar semantics in the embedding space. We therefore adopt a iterative two-stage approach.

1. For some $i \in I_\rho$, retrieve the $k$ nearest candidates $\mathcal{C}(i)$ in embedding space.
2. Select $j^\star(i) = \arg\max_{j \in \mathcal{C}(i)} S(i, j)$ and set $f(i) = j^\star(i)$, $f(j^\star(i)) = i$. Remove both from $I_\rho$.
3. After traversal, pair any remaining tokens in $I_\rho$ at random.

Using approximate nearest neighbors for candidate retrieval, the solver runs in time $O\big(nk(\ell^2 + d + \log n)\big)$ and memory $O(n + nk)$, versus $O(n^3)$ time and $O(n^2)$ memory for a global permutation via the Hungarian method Kuhn (1955).[2]

### 3.5 Encryption Adaptation Training (EAT)

Given bijection $f$ and translator $(E_\rho, D_\rho)$, we adapt the target model $\mathcal{M}_{\text{target}}$ to the alien language by API-only fine-tuning on text examples without any access to the model information.

**Data translation.** For a supervised set $\mathcal{D} = \{(x_i, y_i)\}_{i=1}^N$,

$$x_i' = E_\rho(x_i), \qquad y_i' = E_\rho(y_i),$$

and we upload only the text pairs $(x_i', y_i')$ to the API. Since the server tokenizes with the original $\tau_{\text{target}}$, the model internally observes sequences of alien text.

**Objective .** Given $\mathcal{D} = \{(x_i, y_i)\}_{i=1}^N$, $x_i' = E_\rho(x_i)$ and $y_i' = E_\rho(y_i)$, the API-side objective is the standard causal language modeling objective function, $\min_\theta \mathcal{L}_{\text{EAT}}(\theta) = -\sum_{i=1}^N \sum_{t=1}^{|y_i'|} \log p_\theta\big(y_{i,t}' \mid x_i', y_{i,<t}'\big)$. The resulting *AlienLM* understands and solves tasks in the Alien Language.

### 3.6 Inference Protocol

As depicted in Figure 1, authorized users who hold the translator exchange only text ($x \xrightarrow{E_\rho} x' \xrightarrow{\text{API }(\mathcal{M}_{\text{alien}})} \hat{y}' \xrightarrow{D_\rho} \hat{y}$). The client encrypts plaintext x into alien text x' and sends it to the API. The server, using the original tokenizer $\tau_{\text{tgt}}$, processes what appears to be gibberish but is actually valid alien text. The API returns alien text $\hat{y}'$, which the client decrypts back to plaintext $\hat{y}$. Unauthorized observers including the server itself see only the alien text pairs, $x', \hat{y}'$, which exhibit large edit distances from any meaningful text and resist decryption attempts, thereby protecting sensitive data throughout the inference process.

## 4 Experiments

### 4.1 Experimental Setup

**Training** We train on 300K instruction-tuning examples (Xu et al., 2025)[3] and 150K reasoning examples[4], covering coding, math, and general Q&A, which are well-built public sets (Xu et al., 2025).

---

[2]Here $n = |I_\rho|$ is the number of tokens to permute, $k$ is the number of nearest neighbors, $l$ is the average string length for edit distance computation, and $d$ is the embedding dimension.

[3]https://huggingface.co/datasets/Magpie-Align/Magpie-Pro-300K-Filtered

[4]https://huggingface.co/datasets/Magpie-Align/Magpie-Reasoning-V1-150K

Table 1: Main results across four backbones and seven benchmarks (accuracy, %). AVERAGE is an unweighted mean. RATIO is relative to the original model. AlienLM uses API-only fine-tuning (EAT) with $\rho=1$.

| Models | Method | MMLU | ARC-E | ARC-C | HellaS | WinoG | TQA | GSM8K | Average | Ratio |
|---|---|---|---|---|---|---|---|---|---|---|
| LLaMA 3 8B | Original | 67.32 | 84.13 | 59.39 | 57.07 | 74.35 | 35.25 | 75.89 | 64.77 | – |
| | Substitution | 25.18 | 26.39 | 20.56 | 26.66 | 47.59 | 25.83 | 1.21 | 24.77 | 38.25 |
| | SentinelLM | 29.92 | 46.34 | 27.56 | 38.47 | 55.09 | 30.23 | 31.08 | 36.96 | 57.06 |
| | **AlienLM** | **46.56** | **72.14** | **44.28** | **47.86** | **61.48** | **35.01** | **63.08** | **52.92** | **81.70** |
| Qwen 2.5 7B | Original | 73.50 | 83.80 | 57.51 | 59.66 | 63.77 | 47.86 | 73.09 | 65.60 | – |
| | Substitution | 26.82 | 29.08 | 20.22 | 27.02 | 50.20 | 27.78 | 1.44 | 26.08 | 39.76 |
| | SentinelLM | 23.03 | 35.52 | 21.16 | 31.78 | 49.41 | 31.95 | 25.32 | 31.17 | 47.51 |
| | **AlienLM** | **57.87** | **73.11** | **49.23** | **48.43** | **63.69** | **33.78** | **75.21** | **57.33** | **87.40** |
| Qwen 2.5 14B | Original | 78.79 | 90.36 | 71.16 | 71.63 | 73.72 | 55.94 | 72.86 | 73.49 | – |
| | Substitution | 26.56 | 28.79 | 18.17 | 27.15 | 49.41 | 28.27 | 1.52 | 25.70 | 44.82 |
| | SentinelLM | 22.95 | 62.54 | 42.32 | 43.38 | 61.48 | 34.39 | 73.09 | 48.59 | 66.12 |
| | **AlienLM** | **65.39** | **79.21** | **53.16** | **50.53** | **66.46** | **38.92** | **80.67** | **62.05** | **84.43** |
| Gemma 2 9B | Original | 71.89 | 89.35 | 69.20 | 60.74 | 74.59 | 43.82 | 74.83 | 69.20 | – |
| | Substitution | 24.51 | 28.54 | 19.54 | 26.37 | 50.75 | 25.21 | 0.30 | 25.03 | 36.17 |
| | SentinelLM | 45.88 | 61.07 | 41.81 | 45.38 | 58.25 | 33.17 | 65.73 | 50.18 | 72.52 |
| | **AlienLM** | **54.71** | **75.04** | **48.81** | **50.66** | **60.85** | **35.50** | **70.81** | **56.63** | **81.83** |

Target LLMs are LLaMA 3 (Dubey et al., 2024), Qwen 2.5 (Yang et al., 2024), and Gemma 2 (Team et al., 2024). For proxy embeddings, we use the frozen LM head of Qwen 2.5 for LLaMA 3 8B and Gemma 2 9B, and the LM head of LLaMA 3 8B for Qwen 2.5-7B and 14B. All models train for two epochs which was enough to show the saturation. Unless noted, we set the encryption ratio to $\rho = 1$. Unless otherwise specified, experiments default to LLaMA 3 8B as the target model.

**Evaluation** We evaluate on seven standard benchmarks; MMLU (Hendrycks et al., 2021) for broad knowledge, ARC-Easy (ARC-E) and ARC-Challenge (ARC-C) (Clark et al., 2018) for science question answering, HellaSwag (Zellers et al., 2019) for commonsense inference, Wino-Grande (Sakaguchi et al., 2021) for coreference-based reasoning, TruthfulQA (TQA) (Lin et al., 2022) for truthfulness, and GSM8K (Cobbe et al., 2021) for math problem solving. We report the average score across the tasks and a relative RECOVERY RATIO(RR) to the original model as $RR = 100 \times \text{Average}_{\text{method}}/\text{Average}_{\text{original}}$.

### 4.1.1 BASELINES

To assess recovery under black-box API encryption, we compare against two baselines. **Substitution** applies the same bijection as *AlienLM* at inference without EAT. **SentinelLM** (Mishra et al., 2024b) adapts models to encrypted inputs by modifying embeddings and fine-tuning on encrypted data. Since our setting only allows black-box API access, we cannot alter embeddings or architecture. Therefore, we implement a simplified variant that performs bijection and EAT only.

### 4.2 MAIN EXPERIMENTS

**AlienLM consistently outperforms baselines.** As shown in Table 1Across four backbones, *AlienLM* preserves over 81% of the original performance on average, while Substitution and SentinelLM are substantially lower. *AlienLM* is the top privacy-preserving method on every benchmark and target LLMs. The largest margins were on GSM8K with 62 to 79 points over Substitution, and 5 to 50 points over SentinelLM. Averaged over tasks, *AlienLM* improves over Substitution by 28 to 36 points and over SentinelLM by 6 to 26 points. These results indicate that naive substitution under $\rho=1$ is insufficient. Both API-only adaptation (EAT) and a bijection designed for learnability and human opacity are both necessary to recover performance, especially on numerically sensitive reasoning such as GSM8K.

Table 2: Robustness under token-level mapping attacks (using $\mathcal{M}_{\text{alien}}$ weights) and frequency-analysis attacks (using proxy datasets). Entries are success rates (%, lower is better). For $\rho < 1$, the Embedding/LM Head success is dominated by the unencrypted fraction $1 - \rho$.

| Model | Encryption Ratio | Token-level (AlienLM weights) | | | Frequency Analysis (Proxy datasets) | | |
|---|---|---|---|---|---|---|---|
| | | Embedding | LM Head | Contextual | SlimOrca | Tulu3 | OLMo 2 |
| LLaMA 3-8B | 100% | 0.11% | 0.01% | 0.04% | 0.00% | 0.00% | 0.01% |
| | 80% | 20.07% | 19.93% | 0.04% | 0.00% | 0.00% | 0.01% |
| | 60% | 40.02% | 39.83% | 0.04% | 0.00% | 0.00% | 0.01% |
| Qwen 2.5-7B | 100% | 0.03% | 0.00% | 0.01% | 0.00% | 0.00% | 0.01% |
| | 80% | 19.97% | 19.92% | 0.02% | 0.00% | 0.00% | 0.01% |
| | 60% | 39.90% | 39.81% | 0.02% | 0.00% | 0.00% | 0.01% |

## 4.3 ROBUSTNESS TO DECRYPTION ATTACKS

In Table 2, we evaluate two attack scenarios under the weight-private, black-box API constraint, considering that adversaries may access encrypted alien text exchanged during inference and the adapted AlienLM weights.

**Token-level mapping attacks using model weights.** We consider a server-side adversary who attempts to recover the bijection by aligning alien tokens to target tokens through nearest neighbor search in three representation spaces: (i) embedding matrix, (ii) LM head, and (iii) contextual last-layer states. The adversary selects:

$$\hat{v} = \arg \min_{v \in \mathcal{V}_{\text{target}}} d_{\text{sim}}\big(e_{\text{alien}}(v'), e_{\text{tgt}}(v)\big).$$

The top-1 attack success rate remains below 0.11% when $\rho = 1$ across all backbones and representation spaces . For $\rho \leq 1$ , the success rate for embedding and LM head attacks approximates $(1 - \rho)$ due to unencrypted tokens. However, when evaluated exclusively on the encrypted subset $I_\rho$, the success rate remains below 0.11%. Contextual attacks achieve at most 0.04% success rate independent of $\rho$.

**Frequency analysis using proxy corpora.** We examine an external adversary who attempts to reconstruct the substitution map through statistical analysis. The adversary matches token distributions between publicly available corpora and the encrypted text. Experiments using SlimOrca Lian et al. (2023), Tulu 3 Lambert et al. (2025), and OLMo 2 Walsh et al. (2025) as proxy datasets yield success rates below 0.01% for all models and encryption ratios. The combination of subword-level bijections over vocabularies exceeding $10^5$ tokens effectively neutralizes classical frequency analysis.

As a result, **AlienLM** exhibits strong resistance to both weight-based mapping and corpus-driven frequency attacks. Also, reducing $\rho$ primarily increases trivial matches on unencrypted tokens, while encrypted tokens remain effectively unrecoverable.

## 4.4 EFFECT OF ENCRYPTION RATIO ON PERFORMANCE

**Encryption ratio controls privacy-utility balance.** The encryption ratio $\rho \in [0, 1]$ determines the proportion of tokens subject to permutation, as defined in 3.2. Setting $\rho = 1$ encrypts all non-special tokens, while smaller values preserve a fraction $(1 - \rho)$ of the original vocabulary. Our main experiments employ $\rho = 1$ for maximum encryption coverage, accepting the inherent performance cost of permuting vocabularies exceeding $10^5$ tokens. Full per-benchmark numbers are in Appendix Table 7.

Figure 2 evaluates performance at $\rho$ intervals of 0.2 across seven benchmarks. The results show a strong negative correlation (Pearson r=-0.9626), with accuracy improving monotonically as $\rho$ decreases. This trend reflects reduced

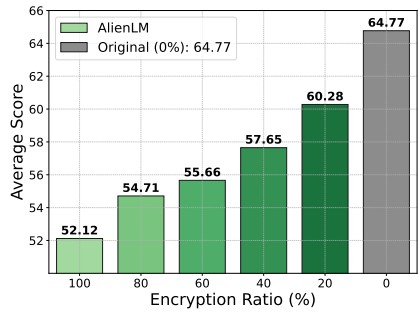

Figure 2: Effect of the encryption ratio $\rho$. Lower $\rho$ permutes fewer token IDs.

lexical distortion at lower encryption ratios, enabling the model to leverage more original linguistic knowledge during adaptation.

The encryption ratio thus provides fine-grained control over the privacy-utility trade-off. While reducing $\rho$ improves performance by preserving more original tokens, the security of encrypted tokens remains uncompromised (see Appendix A.9 for detailed security analysis). This property enables selective encryption strategies where $\rho$ can be tuned based on application requirements, encrypting only sensitive content while maintaining overall utility.

## 4.5 DOMAIN-SPECIFIC FINE-TUNING

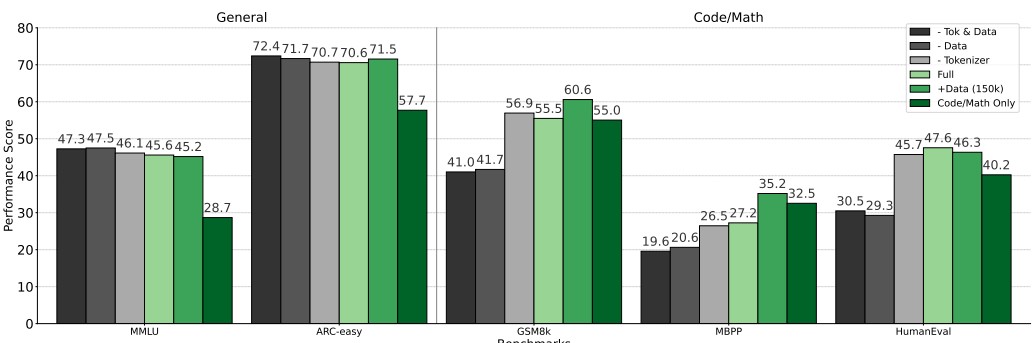

Figure 3: Domain-specific EAT: effect of adding/removing code and math data during EAT, and of excluding those domains from encryption when building the Alien Language.

We investigate domain-specific EAT to understand how *AlienLM* performs when tailored for particular applications such as coding assistants or mathematical reasoning systems. Using domain-annotated Magpie datasets,[5][6] we fix the training size to 300K examples for all experimental conditions, and add an additional 150K domain-specific examples only in the +Data setting. For code evaluation, we additionally report results on MBPP (Austin et al., 2021) and HumanEval (Chen et al., 2021) benchmarks. See appendix tables 8 and 9 for full results.

Figure 3 presents performance across general and code/math benchmarks under five training configurations. The results reveal several key insights. First, excluding code or math data from EAT (-Tok & Data) severely degrades performance on the corresponding domain tasks, with MBPP dropping from 35.2% to 19.6% and GSM8K falling from 60.6% to 41.0%. Second, excluding domain-specific tokens during bijection construction (-Tokenizer) while retaining the training data shows negligible impact, suggesting that vocabulary permutation does not inherently harm domain-specific capabilities. Third, augmenting EAT with additional code/math examples (+Data) provides consistent improvements on domain tasks (GSM8K: 55.5% → 60.6%, MBPP: 27.2% → 35.2%) without compromising general performance. However, training exclusively on code/math data (Code/Math Only) yields inferior results compared to augmentation, particularly on general benchmarks where MMLU drops to 28.7%.

These findings demonstrate that AlienLM maintains domain adaptability while preserving general capabilities. The optimal strategy involves training on diverse data with targeted augmentation for specific domains, rather than narrow specialization that sacrifices broader utility.

## 4.6 SEED-BASED BIJECTION DIVERSITY

A practical encryption system requires generating distinct keys for different users while maintaining consistent performance. We evaluate this property by examining bijection diversity across random seeds using a bucketed pairing strategy, where tokens are grouped before applying the greedy algorithm.

---

[5]https://huggingface.co/datasets/Magpie-Align/Magpie-Llama-3.1-Pro-300K-Filtered

[6]https://huggingface.co/datasets/Magpie-Align/Magpie-Llama-3.3-Pro-500K-Filtered

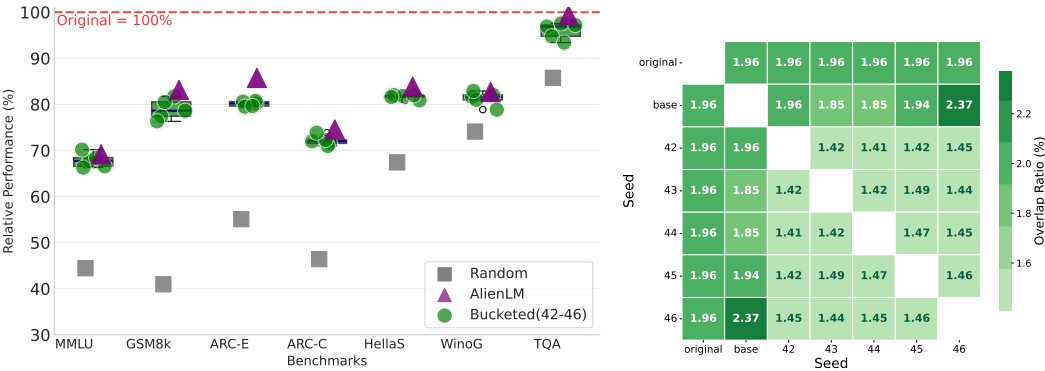

(a) Average accuracy across seeds with bucketed-bijection EAT.    (b) Token-overlap heatmap between seeds.

Figure 4: Tokenizer robustness and key diversification under different random seeds. (a) Utility impact of bucketed greedy search vs. global greedy that optimizes Eq. equation 2. (b) Pairwise overlap of encrypted token mappings.

Figure 4a compares three approaches: random permutation, global greedy AlienLM, and bucketed greedy with multiple seeds (42-46). The bucketed approach achieves comparable performance to global greedy optimization despite potential local optima. Across five random seeds, performance variance remains minimal (Var=0.978 on seven-benchmark average), with all seeds maintaining over 78% relative performance on most benchmarks. The slight performance gap between bucketed and global greedy methods is offset by computational efficiency and consistent results across initializations.

Analysis of bijection overlap reveals that different seeds generate highly distinct mappings, with a maximum pairwise overlap of only 1.96% . This low overlap demonstrates that our framework naturally supports key diversification—each user can obtain a unique Alien Language by simply varying the random seed, analogous to generating distinct cryptographic keys. The combination of performance stability and bijection diversity enables practical deployment scenarios where multiple users require independent encryption schemes without compromising utility.

## 5    CONCLUSION

We presented *AlienLM*, a framework that reinterprets encryption as language translation for weight-private, black-box API LLMs. Using only public tokenizers and vocabularies, *AlienLM* constructs an Alien Language via a vocabulary-level bijection and adapts models through API-only fine-tuning, yielding a lossless client-side translator while preserving model utility. Empirically, across four LLMs and seven benchmarks, *AlienLM* preserves over 81% of the original performance, substantially outperforming substitution- and obfuscation-based baselines.

Decryption attack analyses show strong robustness against weight-based token-mapping and corpus-driven frequency attacks. Beyond aggregate recovery ratio, *AlienLM* exposes operational control key for deployment: a tunable encryption ratio $\rho$ enables fine-grained privacy–utility trade-offs. Also, domain-specific EAT improves math/code performance without harming general capability, and seed-driven bijection diversity provides natural key diversification. Together, these results demonstrate that encryption-as-language can be deployed as a low-overhead, drop-in layer.

While effective, our bijection solver is heuristic, and future work remains. Directions include formalizing the learnability–opacity trade-off under adaptive adversaries, developing stronger global or differentiable solvers, scheduling $\rho$ at the span or content level, incorporating context-aware alien language translation, and integrating with complementary protections such as DP, FL, or TEEs alongside practical key management. By elevating encryption to a language abstraction that LLMs can natively acquire, *AlienLM* opens a practical and extensible path toward composable, privacy-preserving LLM systems.

ETHICS STATEMENT

This work focuses on developing a privacy-preserving framework for API-based LLMs and does not involve human subjects, personal data collection, or deployment of systems with direct social impact. All experiments rely on publicly available benchmarks (e.g., MMLU, GSM8K, HellaSwag) and open datasets such as Magpie, which are widely used in the research community. While our method is designed to strengthen data security by preventing unauthorized access to sensitive prompts or outputs, we acknowledge that any cryptographic mechanism may also be misused if applied maliciously. We therefore release our work strictly for research purposes and emphasize responsible use in accordance with the ICLR Code of Ethics.

REPRODUCIBILITY STATEMENT

We have made extensive efforts to ensure reproducibility. Details of our training procedure, datasets, and evaluation setup are provided in the main text (Section 4) and Appendix A.2–A.3. Hyperparameters for Encryption Adaptation Training (EAT) and vocabulary bijection construction are listed in tables, while pseudocode for the solver and translator is included in Appendix A.7 and A.8. The compute environment is reported in Appendix A.4, and full ablation and robustness analyses are presented in Appendix A.5–A.11. All datasets used are public and properly cited in Section 4. In line with ICLR guidelines, these references to the main text and appendix collectively enable independent researchers to reproduce our results. We further commit to releasing our implementation and scripts in an open-source repository upon camera-ready submission.

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

# A APPENDIX

## A.1 REGULATORY BACKGROUND

A variety of regulatory frameworks explicitly mandate or encourage encryption as a technical safeguard for sensitive data.

- In the EU, the General Data Protection Regulation (GDPR) explicitly lists encryption as an appropriate technical measure, and Network and Information Security 2 (NIS2) strengthens controls for essential services (European Union, 2016; 2022).

- In North America, the National Institute of Standards and Technology (NIST) provides cryptographic and key-management guidance (Barker, 2020), while sectoral rules such as the Health Insurance Portability and Accountability Act (HIPAA) and the Gramm-Leach-Bliley Act (GLBA) require the protection of data in transit and at rest (U.S. Department of Health and Human Services, 2003; Federal Trade Commission, 2002).

- In Asia, examples include Korea's Personal Information Protection Act (PIPA), Japan's APPI (Japan Act on the Protection of Personal Information), Singapore's Personal Data Protection Commission (PDPA), and India's Digital Personal Data Protection Rules Act (DPDP) (of Korea, 2023; Japan, 2022; Singapore, 2022; india, 2025).

These regulations collectively highlight the global importance of encryption as a safeguard for LLM deployment.

## A.2 EAT (ENCRYPTION ADAPTATION TRAINING) HYPERPARAMETERS

Table 3: Training hyperparameters used for EAT across all backbones unless otherwise noted.

| Setting | Value |
|---|---|
| Global batch size | 8 |
| Gradient accumulation steps | 4 |
| Local batch size | 2 |
| Max sequence length | 2048 |
| Optimizer | Paged AdamW (8-bit) Dettmers et al. (2022) |
| Learning rate schedule | Constant |
| Learning rate | 2e-5 |
| Sample packing | True |
| Mixed precision | bfloat16 |

**Notes.** The effective global batch size is computed as local_bsz $\times$ grad_acc $\times$ #GPUs $= 2 \times 4 \times 1 = 8$. We enable sample packing to reduce padding overhead at a fixed maximum length of 2048 tokens. Mixed precision training with `bf16` improves memory efficiency without numerical instability.

## A.3 BIJECTION (ALIEN LANGUAGE) HYPERPARAMETERS

Table 4: Hyperparameters for the vocabulary bijection optimization.

| Setting | Value |
|---|---|
| Levenshtein (edit) weight, $w_{\text{lev}}$ | 1 |
| Embedding-similarity weight, $w_{\text{sim}}$ | 0.01 |
| pairing batch size ($B$) | 50 |
| # nearest neighbors ($k$) | 50 |

**Scoring.** We use the pair score $S(i,j) = -w_{\text{lev}} \cdot \tilde{d}_{\text{edit}}(S(i), S(j)) + w_{\text{sim}} \cdot \text{sim}(e(i), e(j))$ with length-normalized edit distance $\tilde{d}_{\text{edit}}$ and cosine similarity on L2-normalized embeddings. Nearest-neighbor candidate reduction is performed with $k=50$, followed by greedy symmetric pairing in batches of $B=50$. In practice, the optimization successfully matches nearly all tokens, with only about 1,000 tokens left unmatched.

## A.4 COMPUTE ENVIRONMENT

Table 5: Hardware.

| Component | Spec / Notes |
|---|---|
| GPU | NVIDIA A100 80GB $\times$ 1 |
| CPU | AMD EPYC 7763 (64-Core) |
| Alien language build time | $\leq$ 20 minutes (on the above machine) |

## A.5 ABLATION STUDY

Table 6: Ablation results on LLaMA 3 8B (accuracy, %). AVERAGE is the unweighted mean over seven benchmarks. AlienLM rows vary components used to construct the bijection or adapt the model. [†]Uses proxy model head $e_P$ under the black-box constraint.

| Methods | Components | MMLU | ARC-E | ARC-C | HellaS | WinoG | TQA | GSM8K | Average |
|---|---|---|---|---|---|---|---|---|---|
| LLaMA 3 | Original | 67.32 | 84.13 | 59.39 | 57.07 | 74.35 | 35.25 | 75.89 | 64.77 |
| 8B | SFT | 63.74 | 80.56 | 53.67 | 53.70 | 71.74 | 37.58 | 76.12 | 62.44 |
| AlienLM | $e_P$ LM Head[†] | 49.42 | 72.14 | **44.28** | 47.86 | 61.48 | 35.01 | 63.08 | 53.32 |
| | $e_{\text{tgt}}$ LM Head | **51.60** | **73.73** | 44.20 | **48.38** | **65.11** | **36.96** | **65.50** | **55.07** |
| | $e_{\text{tgt}}$ Embeddings | 50.82 | 68.64 | 43.67 | 47.98 | 64.01 | 36.47 | 64.14 | 53.68 |
| | Random $\mathcal{V}$ | 29.92 | 46.34 | 27.56 | 38.47 | 55.09 | 30.23 | 31.08 | 36.96 |

Table 6 reports ablations that isolate the role of proxy versus target representations, as well as the effect of random substitution. We highlight three observations. Firstly, using the target LM head, $e_{\text{tgt}}$, yields the highest average accuracy (55.07%), but the improvement over the proxy LM head, $e_P$, (53.32%) is modest (+1.75 points), indicating that proxy embeddings capture sufficient similarity structure for effective bijection construction in the black-box setting. The relative gains are most evident on WinoGrande (+3.63) and GSM8K (+2.42). Second, substituting the target embedding matrix for the LM head leads to smaller and less consistent improvements (53.68%). Lastly, random vocabulary permutation results in severe degradation (36.96% average, 31.08% on GSM8K), confirming that naive substitution without bijection optimization fails to maintain utility under encryption.

These ablations demonstrate that (a) the proposed proxy-based approach achieves near-optimal performance without access to target internals, supporting its practicality for black-box APIs, and (b) principled bijection design is indispensable; mere random substitution catastrophically undermines task performance.

## A.6 TIME AND MEMORY COMPLEXITY

### A.6.1 TIME COMPLEXITY OF THE BIJECTION SOLVER

Let $n = |I_\rho|$ denote the number of tokens to permute, $d$ the embedding dimension, and $\ell$ the average string length for edit distance. The algorithm consists of three main components:

**1. Candidate retrieval.** For each token $i \in I_\rho$, we query an approximate nearest neighbor (ANN) index to obtain the top-$k$ embedding neighbors. Building the ANN index requires $O(nd)$ time and $O(nd)$ space. Each query runs in $O(k \log n)$ time.[7] Thus, retrieving neighbors for all $n$ tokens costs

$$O(n \cdot k \log n).$$

**2. Pair scoring.** For every candidate pair $(i, j)$, we compute the score..

$$S(i, j) = -w_{\text{lev}} \cdot \tilde{d}_{\text{edit}}(S(i), S(j)) + w_{\text{sim}} \cdot \text{sim}(e(i), e(j)).$$

- Edit distance: $O(\ell^2)$ for two strings of length $\ell$.

---

[7]For FAISS (Johnson et al., 2019) or HNSW-based indices, the empirical complexity scales as $O(\log n)$ per neighbor.

- Cosine similarity: $O(d)$ on L2-normalized embeddings.

Since each token considers $k$ candidates, the total scoring cost is

$$O\big(n \cdot k \cdot (\ell^2 + d)\big).$$

**3. Greedy pairing.** After scoring, tokens are greedily paired with their highest-scoring candidate. Each token is removed once paired, so the overall greedy traversal requires $O(n)$ additional steps.

**Total complexity.** Combining the three components yields

$$O\big(nk(\ell^2 + d + \log n)\big).$$

The memory complexity is $O(n + nk)$ for storing the index and candidate sets.

**Comparison to global matching.** For reference, solving the bijection as a maximum-weight perfect matching with the Hungarian algorithm would require $O(n^3)$ time and $O(n^2)$ memory, which is intractable for vocabularies of size $n \approx 10^5$. Our $k$-NN + greedy solver therefore provides a scalable approximation that runs within minutes in practice.

## A.7 PSEUDOCODE FOR BIJECTION CONSTRUCTION

**Scoring (concept).** We define the pairwise score as a trade-off between human opacity and LLM learnability:

$$S(i,j) = -w_{\text{lev}} \cdot \tilde{d}_{\text{edit}}\big(S(i), S(j)\big) \;+\; w_{\text{cosine}} \cdot \cos\big(e(i), e(j)\big),$$
$$\text{where } w_{\text{lev}} + w_{\text{cosine}} = 1$$

where $\tilde{d}_{\text{edit}}$ is the length-normalized edit distance between token strings and $\cos$ denotes cosine similarity on L2-normalized embeddings.

## A.8 TRANSLATOR (PSEUDO-CODE)

We compose the original tokenizer $\tau_{\text{tgt}}$ and an alien tokenizer $\tau_{\text{alien}}$ induced by the permuted vocabulary. The translator exposes: `encode`: plaintext $\to$ alien text, and `decode`: alien text $\to$ plaintext. This realizes the formal definition in Section 3.2.

---

**Algorithm 2:** Translator using original & alien tokenizers

---

1 **Function** ENCODE($x$)
2    $i \leftarrow \tau_{\text{tgt}}(x)$          // plaintext $\to$ target IDs
3    $x' \leftarrow \tau_{\text{alien}}^{-1}(i)$          // IDs $\to$ alien text
4    **return** $x'$

5 **Function** DECODE($x'$)
6    $i' \leftarrow \tau_{\text{alien}}(x')$          // alien text $\to$ alien IDs
7    $x \leftarrow \tau_{\text{tgt}}^{-1}(i')$          // IDs $\to$ plaintext
8    **return** $x$

9 **Notes:** Special-token set $\mathcal{S}$ is excluded from the permutation; both tokenizers share the same ID space up to the bijection, ensuring $D_\rho(E_\rho(x)) = x$.

---

## A.9 ENCRYPTION RATIO: FULL RESULTS

## A.10 DOMAIN-SPECIFIC FINE-TUNING: FULL RESULTS

**Setup.** Table 8 varies tokenizer scope and training data composition while keeping all other settings fixed. Table 9 reports the corresponding code-specific benchmarks .

---

**Algorithm 1:** Approximate Bijection via kNN Candidate Reduction and Greedy Pairing

---

**Input** : $f$: bijection that maps natural language ID set into Alien Id set; $I_\rho$: token ID set to permute; $S(i)$: surface string; $e(i) \in \mathbb{R}^d$: proxy/target embeddings; $k$: #neighbors, $B$: batch size, $\texttt{lev\_w} \in [0,1]$, $\texttt{sim\_w} = 1 - \texttt{lev\_w}$

**Output**: Permutation $f : I_\rho \rightarrow I_\rho$ (bijection)

**1** **Indexing.** Build FAISS (inner-product) index on L2-normalized matrix $X = [e(i)]_{i \in I_\rho}$.

**2** $Available \leftarrow I_\rho$,

**3** $Pairs \leftarrow \emptyset$.

**4** **foreach** *batch $I_b \subset I_\rho$ of size $B$* **do**

**5**     $I_b \leftarrow I_b \cap Available$; **if** $I_b = \emptyset$ **then**

**6**        continue

**7**     Query index with $Q = [e(i)]_{i \in I_b}$ to get (*Sims*, *NbrIdx*) of top-$k$ neighbors.

**8**     **foreach** $i \in I_b$ **do**

**9**        **if** $i \notin Available$ **then** continue

**10**        Initialize candidate set $\mathcal{C} \leftarrow \emptyset$.

**11**        **for** $c = 1$ **to** $k$ **do**

**12**           $j \leftarrow$ ID from *NbrIdx*$[i, c]$.

**13**           **if** $j \in Available$ **and** $j \neq i$ **then**

**14**              $d_{\text{edit}} \leftarrow \frac{\text{EditDistance}(\text{strip}(S(i)), \text{strip}(S(j)))}{\max(|S(i)|, |S(j)|)}$.

**15**              $sim \leftarrow Sims[i, c]$.

**16**              $score \leftarrow -\texttt{lev\_w} \cdot d_{\text{edit}} + \texttt{sim\_w} \cdot sim$.

**17**              $\mathcal{C} \leftarrow \mathcal{C} \cup \{(j, score)\}$.

**18**        **if** $\mathcal{C} \neq \emptyset$ **then**

**19**           $(j^\star, s^\star) \leftarrow \arg\max_{(j, score) \in \mathcal{C}} score$.

**20**           $Pairs \leftarrow Pairs \cup \{(i, j^\star, s^\star)\}$;    $Available \leftarrow Available \setminus \{i, j^\star\}$.

**21** **Fallback pairing.** Randomly pair remaining IDs in $Available$ and append to $Pairs$.

**22** **foreach** $(i, j, score) \in Pairs$ **do**

**23**     **if** $i \notin \text{dom}(f)$ **and** $j \notin \text{rng}(f)$ **then**

**24**        $f(i) \leftarrow j$,    $f(j) \leftarrow i$.

**25** **return** $f$.

---

Table 7: Effect of the encryption ratio $\rho$ on benchmark. Average is the unweighted mean; Ratio is relative to the original model.

| Method | Ratio (%) | MMLU | ARC-E | ARC-C | HellaS | WinoG | TQA | GSM8K | Average |
|---|---|---|---|---|---|---|---|---|---|
| Original | 100 | 67.32 | 84.13 | 59.39 | 57.07 | 74.35 | 35.25 | 75.89 | 64.77 |
| AlienLM ($\rho$=0.2) | 93.06 | 60.18 | 77.61 | 52.05 | 53.32 | 70.01 | 37.58 | 71.19 | 60.28 |
| AlienLM ($\rho$=0.3) | 89.01 | 57.31 | 76.01 | 47.44 | 51.63 | 66.38 | 34.76 | 70.05 | 57.65 |
| AlienLM ($\rho$=0.6) | 85.93 | 53.98 | 74.33 | 44.62 | 49.70 | 65.43 | 35.74 | 65.81 | 55.66 |
| AlienLM ($\rho$=0.8) | 84.46 | 51.98 | 73.70 | 44.54 | 48.96 | 63.14 | 34.52 | 66.11 | 54.71 |
| AlienLM ($\rho$=1) | 82.33 | 49.42 | 72.14 | 44.28 | 47.86 | 61.48 | 35.01 | 63.08 | 53.32 |

**Findings on general capability.** Excluding domain data (-data) reduces the seven-task average from 47.56 to 29.27 (–18.29), with GSM8K dropping from 55.50 to 41.70 (–13.80). Further excluding domain tokens from permutation as well (-tok & data) is similarly poor (30.49).

Compared to FULL, (-tokenizer) where domain data kept, but domain tokens not permuted, yields a comparable average (45.73 vs. 47.56), indicating that vocabulary permutation itself is not the dominant factor for general capability when training data cover the domain.

Table 8: Domain-specific fine-tuning on general benchmarks (LLaMA3-8B). AVERAGE is over MMLU, ARC-E, ARC-C, HellaSwag, WinoGrande, TruthfulQA, GSM8K.

| Models | Method | Tokenizer | Data | MMLU | ARC-E | ARC-C | HellaS | WinoG | TQA | GSM8K | Average |
|--------|--------|-----------|------|------|-------|-------|--------|-------|-----|-------|---------|
| LLaMA3-8B | full | O | O | 45.59 | 70.58 | 42.41 | 47.32 | 61.25 | 31.21 | 55.50 | 47.56 |
| | - tokenizer | X | O | 46.13 | 70.71 | 41.89 | 47.62 | 58.96 | 31.82 | 56.94 | 45.73 |
| | - data | O | X | 47.50 | 71.68 | 42.06 | 47.86 | 61.56 | 31.95 | 41.70 | 29.27 |
| | - tok & data | X | X | 47.26 | 72.39 | 43.26 | 47.72 | 59.27 | 33.17 | 41.02 | 30.49 |
| | code/math only | only | only | 28.68 | 57.70 | 35.84 | 40.71 | 56.20 | 34.88 | 55.04 | 40.24 |
| | + data | O | +150k | 45.18 | 71.55 | 43.09 | 48.15 | 62.75 | 32.80 | 60.60 | 46.34 |

Table 9: Domain-specific fine-tuning on code/math benchmarks (LLaMA3-8B). AVERAGE is over MBPP and HumanEval.

| Models | Method | Tokenizer | Data | MBPP | HumanEval | Average (Code) |
|--------|--------|-----------|------|------|-----------|----------------|
| LLaMA3-8B | full | O | O | 27.25 | 23.28 | 25.27 |
| | - tokenizer | X | O | 26.46 | 21.96 | 24.21 |
| | - data | O | X | 20.63 | 16.14 | 18.39 |
| | - tok & data | X | X | 19.58 | 15.34 | 17.46 |
| | code/math only | only | only | 32.54 | 26.46 | 29.50 |
| | + data | O | +150k | 35.19 | 29.89 | 32.54 |

**Findings on code/math capability.** Training only on code/math (code/math only) improves code average from 25.27 to 29.50 (+4.23) but performs poorly on general tasks (40.24 average in Table 8).

Adding +150k domain examples (+data) substantially boosts code average from 25.27 to 32.54 (+7.27) and improves GSM8K from 55.50 to 60.60 (+5.10), while keeping general performance in a similar range (46.34 vs. 47.56).

As result, we can conclude into some aspects. (a) Data coverage drives domain competence: including domain data during EAT is crucial; tokenizer-side decisions (permuting vs. exempting domain tokens) are secondary for utility. (b) Augment, don't silo: targeted domain augmentation recovers (and often improves) code/math performance without sacrificing broad competencies, whereas domain-only training trades off generality for smaller gains. (c) Operational guidance: for deployments prioritizing code/math, prefer full or +data with diverse training corpora; consider -tokenizer only when operational constraints require exempting domain tokens from permutation.

## A.11 SEED DIVERSITY AND ROBUSTNESS

Table 10: Performance of LLaMA3-8B under different bijection strategies. Average is the unweighted mean over all benchmarks; Ratio is relative to the original model. Random-5-seed results show robustness and diversity across initializations.

| Models | Method | Average | Ratio | MMLU | ARC-E | ARC-C | HellaSwag | WinoG | TQA | GSM8K |
|--------|--------|---------|-------|------|-------|-------|-----------|-------|-----|-------|
| LLaMA3-8B | Original | 64.77 | – | 67.32 | 84.13 | 59.39 | 57.07 | 74.35 | 35.25 | 75.89 |
| | Random | 36.96 | 57.06 | 29.92 | 46.34 | 27.56 | 38.47 | 55.09 | 30.23 | 31.08 |
| | AlienLM-Magpie | 52.92 | 81.70 | 46.56 | 72.14 | 44.28 | 47.86 | 61.48 | 35.01 | 63.08 |
| | *bucketed pairing* | | | | | | | | | |
| | seed=42 | 50.98 | 78.71 | 45.59 | 67.47 | 42.49 | 46.80 | 60.69 | 34.15 | 59.67 |
| | seed=43 | 51.16 | 78.98 | 44.82 | 67.93 | 42.75 | 46.15 | 61.01 | 33.41 | 62.02 |
| | seed=44 | 50.45 | 77.89 | 44.61 | 67.80 | 42.15 | 46.68 | 60.22 | 32.93 | 58.76 |
| | seed=45 | 51.57 | 79.61 | 47.24 | 66.88 | 42.92 | 46.78 | 61.64 | 34.39 | 61.11 |
| | seed=46 | 50.62 | 78.15 | 46.07 | 67.00 | 43.86 | 46.57 | 58.64 | 34.27 | 57.92 |
| | **Mean** | 51.28 | 79.17 | 45.82 | 68.20 | 43.08 | 46.81 | 60.61 | 34.03 | 60.43 |
| | **Std** | 0.89 | 1.38 | 1.01 | 1.97 | 0.82 | 0.57 | 1.10 | 0.74 | 1.98 |

**Findings.** Table 10 compares different bijection strategies and reports the effect of random seeds under the bucketed greedy solver. Across five random seeds, the mean performance is 51.28% with a standard deviation of only 0.89, corresponding to less than ±2 points on any benchmark. This indicates that bucketed greedy search yields stable performance despite the stochastic initialization. Also, Performance under random-seed bucketed pairing is close to the global greedy AlienLM baseline (52.92%), demonstrating that the bucketed approach remains competitive while being more efficient to compute. All seed-based AlienLM variants substantially outperform naive random vo-

cabulary swap (36.96%), confirming that the optimization objective consistently preserves utility regardless of initialization.

These results suggest that seed variation provides a natural mechanism for key diversification: different seeds lead to distinct bijections with low overlap, while maintaining similar performance levels. From a empirical perspective, this enables issuing multiple encryption keys across users without compromising utility, while also enhancing security by preventing reuse of a single deterministic mapping.

## A.12 QUALITATIVE EXAMPLES (NATURAL VS. ALIEN)

We provide qualitative examples of Alien Language outputs across different tasks including math, coding and creative writing.

Since our tokenizer is based on Byte-Pair Encoding (BPE) (Sennrich et al., 2016), some tokens correspond to non-printable byte sequences. These tokens are omitted from the figures for readability. In addition, when a token string exists but cannot be rendered in LATEX due to Unicode limitations, we display it as `<<UNICODE>>`.

---

**GSM8K**

> **Natural Language**
>
> **Q:** Janet's ducks lay 16 eggs per day.  She eats three for breakfast every morning and bakes muffins for her friends every day with four.  She sells the remainder at the farmers' market daily for \$2 per fresh duck egg.  How much in dollars does she make every day at the farmers' market?
> **A:**"Janet lays 16 eggs per day.  She eats 3 and bakes 4. So she has 16 – 3 – 4 = 9 eggs left.  She sells 9 eggs for 2 dollars each.  9 x 2 = 18.  The answer is 18.

> **Alien Language**
>
> **Q:** Feb Beetle|||||||||||||||| ck anlayış 116 gs perpetual Everyday, Cats five withreak ning but Ib tweaking puff insulting with girlfriends Everyday for six, dell aremainder zatím a Bers' supermarketsdaily with 212 perpetual freshwaterduckegg, HOW hodin jars doesnt Everyday zatím a Bers' supermarkets!
> **A:** Jan overlays116 gs perpetual Everyday,Cats 433 but Ib tweaking 344, Sox 116 gs right,dell 998 gs with 212 jars , 998 data 212 181, A answered 181;

---

In math problems, we observe partial semantic preservation: numbers are often mapped to other numbers of similar scale (e.g., "16" $\rightarrow$ "116"), and ingredients such as "eggs" are mapped to semantically related terms like "jars."  While such mappings retain fragments of meaning, they ultimately obscure full sentence interpretation.  The numeric consistency likely facilitates model-side learnability and helps explain why *AlienLM* substantially outperforms random permutation in reasoning-heavy tasks.

---

MBPP

**Natural Language**

Q: Write a python function to reverse only the vowels of a given string.
A:

```python
def reverse_vowels(s):
    vowels = 'aeiouAEIOU'
    s_list = list(s)
    left, right = 0, len(s) - 1
    while left < right:
        if s_list[left] not in vowels:
            left += 1
        elif s_list[right] not in vowels:
            right -= 1
        else:
            s_list[left], s_list[right] =
            s_list[right], s_list[left]
            left, right = left + 1, right - 1
    return ''.join(s_list)
```

**Alien Language**

Q: Read the/python functionality reverse a towels the stringByAppendingString,
A: getopt(t productList]tyleAttrreverse_w blowjob Michaels(t":  towels "<<UNICODE>> iNdEx<<UNICODE>>IOExceptionZa" helps productList (t] right.  left 100.  düzenlenen(t]  211 right left?  helps productList[top) hodin towels?  right += 211 felse helps productList[mid) hodin towels?  leftsdale 211 elsewhere? helps productList[top), helps productList[mid) helps productList[mid), helps productList[top} right.  left right % 211.  left / 211

---

For code, these tendencies become more pronounced. Some variable names such as `productList` remain consistent across occurrences, allowing the model to maintain syntactic coherence. However, a large fraction of tokens are mapped at the byte level, producing unreadable identifiers. This preserves token-level consistency for the model while making the code nearly impossible to interpret for humans.

**Creative Generation**

**Natural Language**

**Q:** Explain the theory of everything to a 6 year old
**A:** What a fun question!
Okay, little buddy, so you know how we can see lots of different things in the world, like animals, plants, and even stars in the sky?
Well, some really smart people have been trying to figure out how all of these things are connected and how they work. They want to know the secret recipe that makes everything in the universe work together.
This secret recipe is called the "theory of everything." It's like a big puzzle, and scientists are trying to solve it by understanding how all the tiny building blocks of the universe, like atoms and particles, interact with each other.
Imagine you have a big box of Legos, and each Lego is like an atom. Just like how Legos can be connected in different ways to make different things, like a castle or a car, atoms can be connected in different ways to make different things, like water or air or even you!
The theory of everything is like finding the special way to connect all the Legos together to make the whole universe work. It's a really hard problem, but scientists are working hard to figure it out.
And the cool thing is, if they can solve this puzzle, they might be able to discover new ways to make things work better, like finding new ways to make energy or heal people. It's like finding the secret to making the most amazing, incredible, and amazing-est thing in the universe!
So, that's the theory of everything in a nutshell! It's a big, exciting puzzle that scientists are working to solve, and who knows, maybe one day you'll be the one to figure it out!

---

**Creative Generation**

```
Alien Language

Q: Excellenceflat atheory anything the 676 month
olduğundan
A: It the functools questionnaire?  Okay.little
muddy.  knowingly Seelots ifferent Clothing hodin a .
Animalia.  transplant.  but starší hodin a skyline!Bien.
really smartphones trying solver standing allergies
atinybuilding blockSize aiverse.  anatom butparticles.
act for ,brakk recipes called a'theory anything," Here'm
the bigotry muzzle.  but capitalists the bigotry Dropbox
Legends osobních.  but ego atom, unjust Legends osobních
linked hodinifferent highways ifferent Clothing.  the
Newcastle the carcinoma.  anatom linked hodinifferent
highways ifferent Clothing.  groundwater impairment
?Intheory anything finding apecial waypoints connectivity
allergies a Legends osobních altogether awholeiverse ,
Here'm thereally hardships woodworking hardships figures
knockout,Or a coolant soothing .  solver muzzle.  ">< 
impeccable discoveries highways Clothing better.  finding
highways zal , Here'm finding a secretive woodworking
solver.  but know.  perhaps Everyday 'd a figures
knockout?  .  'm atheory anything hodin the nut?  Here'm
the bigotry.  citing muzzle capitalists
```

---

In open-ended text generation, the alienized output appears superficially like natural language but is in fact an opaque mixture of multiple languages. For example, "Legos" in the input is mapped to "Legislature osobních" (a Czech-English mixtured phrase meaning "legislature personal"). Such multilingual, fragmented substitutions render human interpretation extremely difficult, even when the text structure looks plausible.

## A.13 LLM USAGE DETAILS

In accordance with the ICLR 2026 policy on large language model (LLM) usage, we disclose that LLMs were employed during the preparation of this paper. Specifically:

- Writing polish: LLMs (e.g., ChatGPT) were used to refine the clarity, grammar, and readability of the manuscript. Substantive intellectual contributions, including experimental design, theoretical analysis, and interpretation of results, were conducted entirely by the authors.
- Literature discovery: LLMs were occasionally used as an aid in identifying relevant related work, after which all references were manually verified and cross-checked by the authors.

No parts of the reported methodology, experiments, or conclusions were generated by LLMs. All scientific content reflects the authors' own work.

