# OpenReview forum: "AlienLM: Alienization of Language for Privacy-Preserving API Interaction with LLMs"
_ICLR.cc/2026/Conference — ICLR 2026 Conference Desk Rejected Submission_

### Official Review · Reviewer_RnNX · 2025-10-25

**Soundness:** 3
**Presentation:** 2
**Contribution:** 3
**Rating:** 4
**Confidence:** 3

**Summary:**

This paper studies privacy protection during black-box API interactions with LLMs and proposes the AlienLM framework.AlienLM differs from traditional, simple encryption methods by reframing encryption as a language translation task for Large Language Models.
AlienLM introduces a method based on vocabulary-level bijection to construct a secret language that balances confidentiality and LLM learnability. This approach provides a new perspective for integrating sensitive data with API-only fine-tuning scenarios. The method described in the paper achieves over 80% usability while maintaining confidentiality. Their experiments also reveal the trade-off between learnability and opacity.

**Strengths:**

1.	The idea of modeling encryption as language translation (encryption-as-language translation) is interesting, offering a new direction for protecting prompts and outputs in black-box API scenarios.

2.	The paper presents a well-structured and logically consistent methodology. Each component of the proposed framework is motivated by sufficient justifications and accompanied by clear definitions.

3.	The effectiveness of AlienLM is thoroughly discussed. The paper provides an extensive experimental analysis covering multiple benchmarks, various encryption ratios, and domain-specific EAT settings, offering a comprehensive understanding of the model’s behavior under different conditions.

4.	By examining the trade-off between normalized edit distance and similarity-based distance, the paper proposes a novel approach to constructing an aligned language, which is both insightful and valuable as a reference.

**Weaknesses:**

1. My major concern is about the practical usability. As the paper mentioned, AlienLM is proposed for sensitive domains such as healthcare, finance, and education. However, for the healthcare and financial domains, data should be strictly protected and uploading data to a remote LLM is prohibited. How would the author justify such cases?

2.	Some Typos. Typo in line 021, there should be a space between “AlienLM” and “provides”.
Typo in line 239, there should be a space between “target model” and “M_{target}”.
Typo in line 316, there should be a comma between “Table 1” and “Across”.

3.	More discussion on real-world usage could be provided, for example, whether encryption would incur higher inference costs or affect reasoning speed.

4.	The encryption method proposed in the paper still relies on a simple vocabulary mapping. My main concern is whether this approach is easily breakable. For instance, in API-based fine-tuning scenarios, the data provided to the model provider might be sufficient for them to reverse-engineer the encryption.

**Questions:**

1. For the healthcare and financial domains, data should be strictly protected, and uploading data to a remote LLM is prohibited. How would the author justify such cases to use AlienLM?

2. How does the AlienLM differ from previous encryption methods, such as Caesar ciphers?

3. Would the Alien language still be inverted by the trained LLM?

---

> ### Author Response · Authors · 2025-11-21
>
> We thank the reviewer for the thoughtful feedback. Below we respond point-by-point to the questions and concerns, highlighting concrete revisions that we will incorporate into the final version.
>
> For context and consolidated results shared across all reviewers, please refer to our **Global Response to Reviewers**, which provides:
>
> - G1: Conservative reframing and explicit threat model specification
> - G2: Expanded security/decipherment evaluation
> - G3: Practical deployment considerations (overhead, safety)
> - G4: Comprehensive baseline comparisons and novelty clarification
> - G5: Presentation and notation improvements
>
> In this response, we address each question directly and reference relevant sections of the Global Response where applicable.
>
> ### A1.1. Practical Usability in Highly Regulated Domains (Healthcare/Finance)
> ```*Responds to: Weakness #1 and Question #1*```
> We agree that the paper should provide a more detailed explanation of the practical usability in highly regulated domains. AlienLM is proposed for regulated domains because **major frameworks such as GDPR, HIPAA/GLBA, and their Asian counterparts do not universally forbid remote or third-party processing**; instead, they typically allow it under consent, contractual safeguards, and risk assessment, while explicitly requiring strong technical measures—most prominently **encryption/pseudonymisation**—to reduce exposure during transit, storage, and use.
>
> ---
>
> ***A1.1. Regulatory Context and Technical Safeguard Requirements*** We first summarize the common regulatory stance on remote processing. Major frameworks allow it conditionally, while requiring encryption/pseudonymisation to minimize exposure. GDPR Art. 32 lists "pseudonymisation and encryption" as appropriate risk-based safeguards for processing, including processor/third-party settings. This explicitly contemplates scenarios where data is transmitted to external processors with technical protections.
>
> HIPAA and GLBA/Safeguards-Rule guidance strongly emphasize encryption of sensitive data in transit and at rest as a practical compliance expectation for systems that do transmit data to external services. HIPAA explicitly distinguishes de-identified data as outside PHI scope, illustrating a common real-world path to permitted remote processing with additional protections.
>
> These frameworks recognize that **categorical prohibition of all remote processing is often impractical** in modern data ecosystems, and instead mandate risk-based technical safeguards when such processing occurs.
>
> ---
>
> ***A1.2. Practical Deployment Space and Use Cases*** As detailed in **Global Response Section G1**, AlienLM targets the **"permitted-but-risk-sensitive" deployment space** where organizations already use remote LLM APIs legally (e.g., via Business Associate Agreements under HIPAA, regulated vendor contracts in finance, or consented processing under GDPR), yet still face residual exposure risks from API-side logging, operator access, or traffic inspection.
>
> AlienLM adds a **reversible, client-side translation layer** that reduces plaintext visibility to the provider while retaining black-box API compatibility-functioning as an additional technical safeguard that complements contractual protections. Typical use cases include;
>
> 1. **De-identified/pseudonymized data**: Healthcare or financial text that has been de-identified and sent to APIs under approved policies, where additional obfuscation provides defense-in-depth
>
> 2. **Enterprise workflows with policy-permitted API use**: Organizations where external LLM use is approved but requires technical safeguards beyond contractual agreements
>
> 3. **Regulated outsourcing**: Scenarios where technical safeguards are mandated alongside legal protections, and organizations seek practical obfuscation layers that work with black-box APIs
>
> ---
>
> ***A1.3. Clarifying Non-Applicable Cases*** We fully agree that strict no-upload regimes exist (certain hospitals/financial institutions with categorical prohibitions on any external data transmission). AlienLM is not meant to override those constraints; such settings require on-premises deployment or white-box secure-computation solutions.
>
> AlienLM complements rather than replaces such approaches, targeting the substantial real-world deployment space where remote API use is permitted under regulatory frameworks but organizations seek stronger technical protections. We will state this boundary clearly in Sec. 4 and ensure the introduction avoids any implication that AlienLM targets prohibited-transmission environments.

---

> ### Author Response · Authors · 2025-11-21
>
> ### A2. Real-World Serving Cost / Reasoning Speed
> ```*Responds to: Weakness #3*```
>
> We agree this practical aspect deserves explicit discussion. We have measured both translator-side overhead and end-to-end serving costs, and found them negligible. **Please see Global Response Section G3.1 and Tables G3-1 and G3-2** for complete measurements. In summary:
>
> - **Translator overhead**: The translator adds only ~22 MB of CPU memory and sub-millisecond per-sample overhead (0.21 ms for alienization and 0.35 ms for recovery on average)
> - **End-to-end performance**: API throughput and latency remain within measurement noise compared to the original pipeline (62.13s vs. 61.82s latency, 34.15 vs. 34.39 tokens/sec throughput)
>
> The dominant costs in API usage remain network communication and remote model inference rather than local translation, so AlienLM does not introduce meaningful serving-time slowdowns. We will add a short "Real-world deployment" paragraph in Sec. 4 to connect these measurements to the intended API usage scenario and clarify that AlienLM is practical for production deployment.
>
> ---
>
> ### A3. Breakability / Provider Reverse-Engineering / Inversion by Trained LLM
> ```*Responds to: Weakness #4, Question #3*```
>
> We agree that AlienLM does not claim cryptographic secrecy; its guarantees are empirical and threat-model-bounded. To directly test provider-side or eavesdropper-side recovery risks under realistic leakage, we substantially expanded our attack evaluation beyond simple nearest-neighbor alignment. **Please see Global Response Section G2** for comprehensive attack evaluation results across four new attack suites:
>
> - **G2.1**: LLM-based inverse-translation attacks (parallel/non-parallel corpora)
> - **G2.2**: Multilingual MT decoder attacks
> - **G2.3**: Classical n-gram frequency analysis with proxy corpora
> - **G2.4**: N-gram analysis with parallel-corpus leakage (known-plaintext)
>
> Across these stronger attack suites, reconstruction remains negligible within practical leakage regimes:
>
> - **LLM-based attacks** with 20-shot parallel examples achieve only BLEU 1.88–11.56, ROUGE-L below 0.25, and LLM-Judge scores near minimum (1.0–1.7 out of 3.0)
> - **MT-based attacks** achieve BLEU ≈ 11.4, comparable to in-context learning with no substantial advantage
> - **N-gram frequency analysis** with optimal Hungarian matching yields token-level accuracy below 0.22% even with 1,000 parallel pairs exposing ~19,673 tokens
>
> These results indicate that provider-side access to alientext (and limited parallel pairs) is insufficient to recover the full bijection at the tested scales. Even strong LLM/MT decoders fail to produce semantically faithful plaintext, and statistical decipherment cannot extrapolate beyond trivially leaked tokens.
>
> ---
>
> **A3.1. Threat Model Scope and Security Assumptions** Our threat model focuses on **API-side adversaries** (providers, eavesdroppers) who observe alientext traffic but do not compromise the client-side system. This assumption is standard and practical:
>
> - **Client-side security is foundational**: Just as TLS/SSH rely on protecting private keys, and password managers rely on protecting master passwords, AlienLM's security model assumes the client-side translator and bijection seed remain under user control. This is a reasonable assumption for deployed systems where the client runs on trusted infrastructure.
>
> - **Defense-in-depth against API-side exposure**: Our expanded attack evaluation (**Global Response Section G2**) demonstrates that even with API-side observation and realistic leakage (up to 1,000 parallel pairs), bijection recovery remains negligible. This provides strong practical protection against the primary threat: API-side plaintext exposure.
>
> - **Explicit scope**: If an attacker gains full access to the client system (compromising the translator or bijection seed), they can perform chosen-plaintext attacks. However, this represents a fundamentally different threat tier—one that defeats most practical security systems including encryption schemes that rely on key secrecy. Such client compromise is outside our deployment model, which assumes the client environment is adequately secured.
>
> This threat model aligns with real-world API deployment scenarios where organizations control their client infrastructure and seek to reduce exposure risks when interacting with external API providers.

---

> ### Author Response · Authors · 2025-11-21
>
> ### A4. Novelty vs. Classic Substitution Ciphers (e.g., Caesar) + Typos
> ```*Responds to: Weakness #2 and Question #2*```
>
> We agree this distinction should be stated explicitly. Classical substitution ciphers (e.g., Caesar/ROT-style character shifts) apply a fixed deterministic transform without aligning the model to the transformed distribution.
>
> **Please see Global Response Section G4.1** for complete baseline comparisons including ROT13 and naive vocabulary substitution.
>
> ---
>
> **A4.1. Key Distinctions from Classical Substitution Ciphers** AlienLM differs from classical substitution ciphers in two essential ways:
>
> 1. **Principled bijection construction**: AlienLM constructs a vocabulary-level bijection optimized for the opacity–learnability trade-off (edit-distance-based unreadability while preserving embedding similarity), rather than applying a fixed character-level shift
>
> 2. **API-only adaptation**: AlienLM performs API-only adaptation so the black-box target model is explicitly realigned to the alien distribution
>
> Together, these enable a reusable reversible translation-based privacy-preserving layer that preserves >80% utility under practical API constraints.
>
> ---
>
> **A4.2. Empirical Comparison** As shown in **Global Response Table G4.1**:
> - **ROT13** (character-level with adaptation): 36.03% relative performance
> - **vocab-sub** (naive vocabulary substitution without adaptation): 38.25% relative performance
> - **AlienLM** (vocabulary-level bijection with adaptation): 81.70% relative performance
>
> In our experiments, naive Caesar/ROT baselines severely disrupt token neighborhoods and subword structure, causing large utility collapse and remaining far below AlienLM's performance level.
>
> ---
>
> **A4.3. Typo Corrections**
> We acknowledge the reported typos and will fix them in the revision:
> - Line 021: Add space between "AlienLM" and "provides"
> - Line 239: Add space between "target model" and "$M_{\text{target}}$"
> - Line 316: Add comma between "Table 1" and "Across"
>
> We will also conduct an additional proofreading pass for minor formatting issues throughout the paper.
>
> ---
>
> We hope these clarifications address the reviewer's concerns. We will incorporate the deployment-scope clarification, overhead discussion, expanded attack evaluation, baseline distinction, and typo fixes into the final revision.

---

> > ### Comment · Reviewer_RnNX · 2025-11-22
> > **Rebuttal Acknowledgement**
> >
> > Dear Authors,
> >
> > I have read the provided responses and would like to thank your efforts. However, I think theoretical security analysis is needed to show that the bijections are not breakable by the internal models. Additionally, even AlienLM achieves the best ratio with the least utility loss, sacrificing ~10\% acc is still prominent.
> >
> > Therefore, I decide to keep my score.

---

> > > ### Author Response · Authors · 2025-11-27
> > >
> > > We thank the reviewer for the thoughtful follow-up and for highlighting two specific points: (i) whether internal models can break the bijection, and (ii) the practical significance of the ∼10% utility drop.
> > >
> > > On the first point, our view is that even with internal access, the bijection is not explicitly “stored” as a decodable key in the weights. AlienLM trains the API-side model to treat the alien vocabulary as its ground-truth language, while the client-side translator is the only place where the bijection is explicitly instantiated. Because modern LMs are heavily overparameterized and permutation-symmetric over both token indices and hidden units (Thennal et al., 2024). Many different parameterizations implement the same function on alientext. In particular, if $\theta$ is a solution that fits the alien corpus, then applying a consistent permutation to the input embedding matrix and output projection yields a different parameter set $\theta^P$ with the same likelihood and behavior. Thus, internal weights are not uniquely tied to a single labeling of tokens, and any attempt to read off a specific vocabulary-level bijection $f$ from one checkpoint $\theta^\*$ alone faces a fundamental non-identifiability: many distinct bijections and parameterizations are observationally equivalent at the level of alien text distributions.
> > >
> > > Empirically, this perspective is consistent with the expanded attack evaluation in our Global Response (Sec. G2): strong LLM-based inverse-translation attacks, MT-style seq2seq attacks, and n-gram + Hungarian matching (with and without parallel leakage) all fail to recover meaningful plaintext or non-trivial fractions of the bijection at realistic leakage scales (BLEU $\approx$ 2–11, ROUGE-L < 0.25, token-level mapping accuracy < 0.22%). We see this combination of structural non-identifiability and empirical failure of strong internal/LLM-based attacks as a first step toward a more formal treatment of internal attackers.
> > >
> > > Regarding the ∼10% utility gap, we agree this loss is not negligible, and we do not intend to claim “free privacy.” Our goal is to characterize a privacy–utility frontier under realistic API-only constraints. As summarized in Global Response Sec. G4.1, AlienLM retains ~81.7% of the original utility, substantially outperforming other black-box / substitution-style baselines under the same threat model and adaptation pipeline (ROT13 and naive vocabulary substitution at ~36-38% relative performance, SentinelLM-black-box at ~57%), while approaching the white-box upper bound. In the final version, we will clarify this positioning more explicitly.
> > >
> > >
> > > **References:**
> > >
> > >
> > > K, Thennal D, Tim Fischer, and Chris Biemann. “Large Language Models Are Overparameterized Text Encoders.” Proceedings of the 10th Workshop on Representation Learning for NLP (RepL4NLP-2025), Association for Computational Linguistics, 2025, pp. 170–184. DOI: 10.18653/v1/2025.repl4nlp-1.13.

---

### Official Review · Reviewer_uqLQ · 2025-11-01

**Soundness:** 2
**Presentation:** 3
**Contribution:** 2
**Rating:** 4
**Confidence:** 4

**Summary:**

The paper studies LLM text obfuscation inference (which is referred to as encryption/decryption in the submission, though its guarantees are far weaker than cryptographic security, making “obfuscation” a more accurate term here).

In particular, the submission proposes to (1) introduce additional "encryption" (encoder) and "decryption'' (decoder) modules on top of the standard LLM API inference protocol, where the "encryption" module transforms plain text into human-unreadable representations (i.e.,  the “Alien Language”) that remain similar to the original plain text in the embedding space, and the "decryption" module converts the inference results back into readable text; and (2) to make the inference process feasible and meaningful, the LLM itself operates on the "Alien Language" and is fine-tuned (via API adaptation) on paired text data in "Alien Language".

Experiments show that the proposed approach preserves much of the model’s utility while resisting simple recovery attempts. In particular, attacks based on nearest-neighbor alignment between "Alien" and target tokens fail to accurately reconstruct the original text.

**Strengths:**

- The paper is overall well structured, and the results are clearly presented.

- It presents an practically relevant topic and interesting idea, and the approach is reasonable and easy to follow.

- The experiments demonstrate some intriguing results.

**Weaknesses:**

- The threat model is not explicitly defined. While it can be roughly inferred from the context, it remains unclear which specific adversarial settings the proposed method can resist. The current evaluation only considers relatively simple token-level mapping attacks (based on nearest-neighbor search in embedding or representation spaces).
In contrast to true cryptographic or secure multi-party computation schemes, where privacy is maintained even when the encryption and decryption algorithms are publicly known (following Kerckhoffs’ principle), this learning-based obfuscation approach relies heavily on the secrecy of the learned mapping. Thus, it does not offer formal or provable privacy guarantees. I would therefore recommend being more cautious and conservative in framing the method as “privacy-preserving”. The authors should also position it within the broader landscape of existing privacy-preserving inference methods, including cryptographic computation, differential privacy, and syntactic or semantic obfuscation approaches. Given the known fragility of heuristic, non-theoretically grounded obfuscation methods [1], the privacy level here appears closer to PII-scrubbing or surface-level anonymization, which already achieve "near-perfect" results in many cases. Overall, this may limit the method’s real-world value.

[1] Carlini, Nicholas, et al. "Is Private Learning Possible with Instance Encoding?" 2021 IEEE Symposium on Security and Privacy (SP). IEEE, 2021.

- The idea of obfuscation used in this submission, i.e., making representations human-uninterpretable (by maximizing some kinds of distance in data space) yet machine-learnable (by minimizing some kinds of distance in embedding space), has been well established and is not particularly novel in the literature. This somewhat diminishes the method’s technical contribution.

**Questions:**

- The paper seems to confuse the token set $S$ and the index set $I$. For example, in lines 198–199, the operation  $S\cup (I \backslash I_\rho)$ is confusing.

- The subscript notation for the target model is sometimes written as "target" and other times as "tgt." It would be better to make this consistent throughout.

- It would help to elaborate on how the proxy mapping $S(v) = τ_{proxy}(v, V)$ is constructed

- The method section could benefit from a clearer and more detailed explanation of the core design ideas. It would improve clarity to explicitly state the design intent at the beginning of each subsection. For example, in Section 3.3:
It could be made clearer by linking the qualitative goals to quantitative objectives: “unreadable to humans” corresponds to maximizing edit distance, while “learnable by the model” corresponds to constraining embedding similarity.

---

> ### Author Response · Authors · 2025-11-21
>
> We sincerely thank the reviewer for the thoughtful and constructive feedback. Your careful critique of the threat model, framing, and positioning has helped us substantially improve the clarity and rigor of our presentation. Below we respond to the two weaknesses and four questions, highlighting concrete revisions that we will incorporate into the final version.
>
> For context and consolidated results shared across all reviewers, please refer to our **Global Response to Reviewers**, which provides:
>
> - G1: Conservative reframing and explicit threat model specification
> - G2: Expanded security/decipherment evaluation
> - G3: Practical deployment considerations
> - G4: Comprehensive baseline comparisons and novelty clarification
> - G5: Presentation and notation improvements
>
> In this response, we address each point directly and reference relevant sections of the Global Response where applicable.
>
> ---
>
> ### W1. Threat Model, Framing, and Security Scope
> We appreciate the reviewer’s careful critique and agree that our original presentation did not make the security scope and adversary assumptions explicit enough. We do not claim cryptographic secrecy under Kerckhoffs’ principle or IND-CPA/semantic-security definitions. Instead, AlienLM is a deployable, reversible, translation-based obfuscation layer for black-box LLM APIs: it reduces practical plaintext exposure to API-side observers while preserving utility under realistic API constraints. In the revision, we will (i) state an explicit threat model, (ii) consistently reframe “encryption” as translation-based obfuscation, and (iii) position AlienLM conservatively within the broader privacy-preserving inference landscape.
>
> ***W1.1. Explicit Threat Model (In-Scope / Out-of-Scope Adversaries)*** We agree the threat model should not be left implicit. We will add a dedicated Threat Model subsection near the end of the Introduction / start of Sec. 3, specifying the adversaries AlienLM is designed and evaluated for:
>
> 1. Primary in-scope adversary (API-side provider adversary).
> The attacker observes only alientext exchanged with the service (including outputs), but has no access to (a) client-side translator, (b) bijection key/seed, or (c) target model internals/weights.
>
> 2. Extended in-scope adversary (limited known-plaintext leakage).
> The attacker additionally obtains a small number of plaintext–alientext pairs (e.g., via logs or user-shared snippets). Our expanded attacks in W2 explicitly evaluate robustness under this realistic leakage regime.
>
> 3. Upper-bound in-scope adversary (white-box surrogate).
> The attacker obtains a model trained on alientext (AlienLM-adapted local model or surrogate) and attempts bijection recovery using embedding/LM-head or representation-space attacks.
>
> 4. Out of scope. Chosen-plaintext/alientext attacks that require querying or compromising the client-side translator or exposing the bijection key are explicitly out of scope: if the translator/seed is compromised, any deterministic bijection-based scheme becomes vulnerable. Our focus is therefore on API-side adversaries without client compromise, and we will state this limitation clearly.
>
> ---
>
> ***W1.2. Translation-based layer for black-box LLM APIs framing*** We acknowledge the reviewer’s point that calling our mechanism “encryption” can be misread as implying IND-CPA/semantic security. We will revise wording throughout (Abstract, Intro, and Sec. 3) to consistently describe AlienLM as a *translation-based, utility-preserving obfuscation for black-box APIs*, whose protection relies on keeping the bijection key local. This aligns with how AlienLM is actually constructed as a bijective alien language with a client-side translator, while avoiding any claim of formal cryptographic guarantees.
>
> ---
>
> ***W1.3. Kerckhoffs-compliant key secrecy via seed-based bijection diversity*** We agree with the reviewer that privacy should not depend on hiding the algorithm. AlienLM follows this spirit: the bijection construction procedure can be public, while privacy relies on the secret seed that instantiates a particular bijection key.
>
> Empirically, Sec. 4.6 shows that running the same bucketed bijection algorithm with different seeds yields highly distinct Alien Languages (pairwise overlap ≤ 1.96%) while utility remains stable across seeds. This provides practical key diversification or rotation analogous to cryptographic systems, and we will make this Kerckhoffs-aligned interpretation explicit in the Threat Model and Limitations.

---

> ### Author Response · Authors · 2025-11-21
>
> ***W1.4. Positioning in the broader privacy-preserving inference landscape***
> We agree that AlienLM should be positioned more explicitly and conservatively within the broader privacy-preserving inference landscape (e.g., HE/MPC/TEEs, DP/FL, and prior obfuscation). To keep W1 focused on threat-model clarification, we defer this discussion to A2, where we present expanded attacks and refine the related-work framing accordingly.
>
> ---
>
> ***W1.5. Heuristic fragility and practical (non-formal) privacy limits***
> We agree that heuristic obfuscation can be fragile in general, as highlighted by Carlini et al. (2021). Our goal is therefore not to claim formal secrecy, but to empirically quantify practical resistance under the explicit API threat model (A1.1). To directly address this concern, we substantially expanded the attack evaluation already presented in our rebuttal to Reviewer 1: (i) LLM-based inverse-translation attacks under both parallel and non-parallel leakage, (ii) strong multilingual MT decoder attacks, and (iii) classical n-gram frequency decipherment with and without known-plaintext parallel pairs.
>
> - ```Table R1-1```: LLM inverse-translation (Parallel Corpus)
> - ```Table R1-2```: LLM inverse-translation (Non-Parallel Corpus)
> - ```Table R1-3```: MT-based inverse-translation
> - ```Table R1-4```: N-gram frequency analysis w/ proxy corpora
> - ```Table R1-5```: N-gram analysis w/ parallel-corpus leakage
>
> Across these stronger suites, reconstruction remains negligible within realistic leakage regimes: even with modest aligned examples or heavy statistical matching, attackers fail to extrapolate beyond trivially leaked tokens, and full bijection recovery stays near zero. These results do not imply cryptographic security, but they place AlienLM beyond simple “PII-scrubbing-level” collapse under our in-scope API adversaries. We will connect these empirical findings explicitly to the clarified threat model and add a limitations note that AlienLM is a deployable, translation-based privacy-preserving layer, not a substitute for formal confidentiality when such guarantees are required.
>
> ### W2. Contribution / Novelty
> We agree that the high-level opacity–utility goal has precedents. Our contribution is not a new formal privacy notion, but making this goal universally deployable under black-box LLM APIs by formulating and estimating a scalable vocabulary-level bijection that enables end-to-end, reusable translation-based privacy preservation. We summarize the key distinctions below.
>
> ---
>
> ***W2.1. Scope: prior work is instance- or object-specific.*** Prompt Obfuscation (Pape et al., USENIX’25) protects system-prompt IP via per-prompt optimization: it obfuscates a given protected prompt and must be re-run when that prompt changes. AlienLM instead learns one reusable alien language (bijection) and adapts once, supporting arbitrary user inputs/outputs end-to-end.
>
>
> ***W2.2. Access assumptions & domain: prior work does not reach API-only general tasks.***
> CodeCipher (Lin et al., 2024) is code-domain-only and relies on white-box embedding manipulation to learn task-specific mappings. AlienLM targets general natural-language tasks and requires no structural access, operating purely with public vocabularies and API-only EAT—matching commercial black-box constraints.
>
> ***W2.3. Closest API-only prior is heuristic rewriting, not an explicit language.***
> EmojiPrompt (Lin et al., NAACL’25) obfuscates prompts through emoji-based/generative rewriting in a black-box setting. AlienLM instead constructs an explicit bijective subword language over ~10^5 tokens and realigns the model to that distribution via EAT, yielding a reusable and reversible translation-based privacy-preserving layer beyond rewrite heuristics.
>
> ***W2.4. We explicitly formulate and solve bijection estimation.***
> Unlike prior methods that apply obfuscation heuristically, AlienLM casts obfuscation as a bijection estimation problem. We explicitly link the qualitative goals—“unreadable to humans” (maximize surface/edit distance) and “learnable by the model” (constrain embedding similarity)—to a target-embedding–constrained objective (Eqs. 1–2), and provide a k-NN reduction + greedy pairing solver that scales to modern vocabularies ($|I_\rho| ≈ 10^5$), enabling practical end-to-end deployment.

---

> ### Author Response · Authors · 2025-11-21
>
> ### Q1. Notation Confusion Between Token Set and Index Set
>  We agree that although $I_\rho$ is introduced at line 181, its meaning and role are not stated clearly enough, which makes the operations around lines 198–199 confusing. In the revision, we will explicitly clarify that $I_\rho$ denotes the index set over which the bijection is defined and applied.
>
> ### Q2. Inconsistent Subscript Notation (“target” vs. “tgt”)
>  We agree this inconsistency may distract readers. In the revision, we will standardize the subscript notation for the target model to a single form (we will use ```tgt``` throughout) and audit Sec. 3–4 and all equations/tables for full consistency.
>
> ### Q3. Clarifying How the Proxy Mapping is Constructed
>  We agree that although we introduced $S(v) = \tau_\text{proxy}(v, V)$ in line 213, we did not state clearly enough that this step is used to construct a proxy embedding for each target token via a (piece-level) weighted average. In our API-only setting, we cannot access the target embedding table, and target-model tokens do not have a one-to-one correspondence with proxy-model tokens due to tokenizer differences. Therefore, we decompose each target token $v$ into proxy pieces $S(v) = \tau_\text{proxy}(v, V)$and approximate its embedding by length-normalized weighted averaging their proxy embeddings:
>
> $$e_{\text{P}}(v) = \frac{1}{|S(v)|} \sum_{s \in S(v)} e_{\text{proxy}}(s)$$
>
> **Example**. if a target token is “unbelievable” and the proxy tokenizer splits it as ```S(v) = {"un", "believable"}```, then the proxy embedding of $e_{\text{P}}(v)$ is computed by averaging the proxy embeddings of these pieces.
>
> ### Q4. Clarifying Design Intent and Goal–Objective Linkage in Methods
>  We agree that the design intent and its link to the objective could be stated more explicitly. In the revision, we will add a short design-intent paragraph at the start of each subsection in Sec. 3 and make the goal–objective correspondence explicit: “unreadable to humans” maps to maximizing surface/edit-distance terms, while “learnable by the model” maps to preserving proxy/target embedding similarity in Eqs. (1-2).

---

### Official Review · Reviewer_EeK6 · 2025-11-02

**Soundness:** 2
**Presentation:** 2
**Contribution:** 2
**Rating:** 2
**Confidence:** 4

**Summary:**

AlienLM is a framework that conceptualizes encryption as language translation for large language models (LLMs) accessed through black-box APIs. The approach builds an “Alien Language” via a vocabulary-level bijection and employs API-only fine-tuning, enabling encrypted communication with commercial LLMs without access to model internals. Experiments across multiple models and benchmarks demonstrate that AlienLM retains over 81% of the original model’s performance while offering strong resistance to token-mapping and frequency-analysis attacks. The framework provides a deployable, low-overhead method for privacy-preserving applications and highlights the separation between linguistic representation and task competence, paving the way for future research on composable privacy-preserving mechanisms and the learnability–opacity trade-off.

**Strengths:**

1. Bijection-based Encrypted Language Layer:
The authors propose constructing an Alien Language and its translator through a vocabulary-level bijection, enabling bidirectional and lossless conversion between plaintext and encrypted alien text. This mechanism conceals model internals while providing a deployable encryption layer fully compatible with black-box LLM APIs.

2. API-only Adaptation (EAT):
The authors introduce an API-only fine-tuning process, termed Encryption Adaptation Training (EAT), which allows the model to adapt to the Alien Language without accessing internal parameters. This adaptation consistently preserves over 80% of the model’s original performance, resulting in the Alien Language–adapted model, Malien.

3. Domain-specific Adaptation:
The authors extend EAT to domain-focused scenarios, demonstrating that targeted fine-tuning improves performance in specialized areas such as coding and mathematical reasoning, while maintaining strong general-purpose capabilities across diverse benchmarks.

**Weaknesses:**

1. Poor presentation and readability: The overall presentation is difficult to follow. In particular, Section 3 (Method) introduces a large number of mathematical formulations without sufficient explanation or intuitive interpretation, making it challenging for readers to understand the core ideas and methodological flow.

2. Insufficient baselines for comparison: The evaluation includes only one baseline, SentinelLM (Mishra et al., 2024b), which is inadequate for demonstrating the effectiveness of the proposed privacy-preserving mechanism. Additional privacy-related baselines, such as approaches based on secure inference, differential privacy, should be included for a more convincing empirical validation.

3. Lack of ablation study: The paper does not provide an ablation analysis to isolate the contributions of key components (e.g., the bijection design, the encryption ratio ρ, or the API-only fine-tuning step). Such experiments are necessary to clarify which elements of the framework drive the observed performance improvements.

**Questions:**

see Weaknesses

---

> ### Author Response · Authors · 2025-11-21
>
> We thank the reviewer for the careful reading and constructive feedback. Below we respond point-by-point to the three weaknesses, highlighting concrete revisions and experiments that we will incorporate into the final version.
>
> For context and consolidated results shared across all reviewers, please refer to our **Global Response to Reviewers**, which provides:
>
> - G1: Conservative reframing and explicit threat model specification
> - G2: Expanded security/decipherment evaluation
> - G3: Practical deployment considerations (overhead, safety)
> - G4: Comprehensive baseline comparisons (SentinelLM white-box, naive substitution baselines)
> - G5: Presentation and notation improvements
>
> In this response, we address each weakness directly and reference relevant sections of the Global Response where applicable.
>
> ---
>
> ### A1. Improve Presentation and Readability
>
> We agree that Sec. 3 is math-heavy and does not sufficiently highlight the core intuition and end-to-end flow. We will re-structure Sec. 3 to guide readers from intuition to formalism, adding an intuition-first explanation before diving into mathematical formulations. For broader presentation improvements, please also see **Global Response Section G5**.
>
> ---
>
> ***A1.1 Add an Intuition-First Overview in the Main Text*** Before Eqs. (1-2), we will insert a brief "method-in-words" overview and a tiny toy example that makes both the bijection initialization and the encryption step tangible.
>
> **Toy corpus example:** Consider a tiny corpus with three sentences such as "hello world", "world cup comes", "come here". We illustrate how one token (`come`) selects its initial alien partner. For each candidate token, we compute (i) semantic closeness (embed sim / sim dist) and (ii) surface dissimilarity (edit dist), then combine them into a final score:
> | candidate | embed sim | sim dist | edit dist | final score |
> | --------- | --------- | -------- | --------- | ------- |
> | comes     | 0.92      | 0.08     | 1         | 0.84    |
> | hello     | 0.06      | 0.94     | 4         | 2.12    |
> | world     | 0.4       | 0.6      | 4         | 2.8     |
> | cup       | 0.07      | 0.93     | 3         | 1.14    |
> | here      | 0.8       | 0.2      | 3         | 2.6     |
>
> From `come`'s perspective, candidates differ along two axes: meaning similarity and surface difference.
>
> - `comes` is very close in meaning but also almost identical in form (edit dist = 1), so pairing `come` and `comes` provides little alienization, yielding a low final score.
> - `here` is strongly different in form, but its meaning is too far from `come` (higher sim dist), making adaptation harder.
> - `world` offers the best balance: moderate semantic similarity with clear surface dissimilarity (edit dist = 4), which maximizes the final score. Hence `come` pairs with `world` by the argmax rule.
>
> **Toy Alien Translation:** Once this 1:1 bijection map is initialized, encryption is a simple token-wise substitution. For illustration, suppose the initialized bijection includes (come, world), (hello, cup), (comes, here).
>
> Then plaintext is translated by swapping each token with its paired alien token:
> - Plaintext: `hello world`
> - Alien text: `cup come`
>
> We will place this walk-through before Eqs. (1–2) so that readers first see (i) the bijection initialization process and (ii) translation process into alien language from
> original text, before encountering the formal objective.
>
> ---
>
> ***A1.2 Provide a Clearer High-Level Flow in Sec. 3*** Detailed pseudocode for bijection construction and the translator is already provided in Appendix A.7–A.8. Rather than duplicating those low-level details in the main text, we will add a concise, intuition-first end-to-end pipeline summary at the beginning of Sec. 3 and explicitly point readers to A.7–A.8 for full algorithmic specification. This makes the methodological flow easy to grasp before formal derivations.

---

> ### Author Response · Authors · 2025-11-21
>
> ### A2. Baseline Selection Under Black-Box API Constraints
>  We appreciate the suggestion to expand baselines. Our target is client-only privacy under black-box LLM APIs: the server never sees plaintext, and the client interacts only via standard text-in/text-out API calls without any provider-side cooperation (no protocol changes, secure hardware, or server-side noise injection). Under this deployable threat model, we have expanded our baseline comparisons. Please see **Global Response Section G4** for complete baseline comparison results.
>
> **Methods outside our deployable scope:**
>
> - **Secure inference / cryptographic protocols (MPC/HE/TEE)**: These require provider cooperation or architectural control, which is unavailable in our API-only black-box setting. For example, homomorphic encryption requires 1,000–10,000× computational overhead and white-box access to model architecture; MPC protocols require prohibitive communication costs and access to model parameters; TEEs require specialized hardware not offered by major API providers. While these methods offer formal security guarantees, they are not deployable as baselines in our setting.
>
> - **Differential privacy**: DP protects training data or released parameters via training-time noise or output randomization. Since we do not control API training or outputs, DP is orthogonal to our inference-time input-privacy goal and not deployable in our setting.
>
> We will clarify this scope explicitly in the revision.
>
> **Expanded baselines under black-box API constraints:** Accordingly, we expanded baselines to those realizable under the same API interface. Please see **Global Response Table G4.1** for complete results. Our comparison now includes:
>
> - **SentinelLM-WhiteBox (upper bound)**: An idealized ceiling when full white-box access to model internals is available, serving to contextualize best-case utility.
>
> - **SentinelLM (black-box, API-compatible prior)**: The closest prior work. Since its original formulation assumes white-box setup, we implement the fully black-box variant with standard text-in/text-out APIs. This serves as the most direct apples-to-apples prior baseline under our client-only constraints.
>
> - **vocab-sub (naive substitution without adaptation)**: Applies the bijection directly to the frozen original model without any adaptation, isolating the contribution of API-only fine-tuning.
>
> - **ROT13 (Unicode-level minimal baseline with adaptation)**: A minimal black-box baseline operating at the Unicode-character level rather than the tokenizer level. We apply a fixed Unicode ROT13 substitution to inputs and perform the same API-only adaptation pipeline as AlienLM. This isolates the effect of using a non-learned, minimal-unit substitution while keeping the same adaptation process.
>
>
> As detailed in **Global Response Section G4.1**:
>
> 1. **Black-box performance**: AlienLM achieves 81.70% of original utility, substantially outperforming SentinelLM (57.06%) in the same black-box setting.
>
> 2. **Gap to white-box upper bound**: AlienLM recovers approximately 84.82% of SentinelLM-WhiteBox performance, demonstrating that API-only adaptation can preserve most utility despite lacking access to model parameters and gradients.
>
> 3. **Necessity of proper adaptation**: Both vocab-sub (38.25%) and ROT13 (36.03%) collapse, highlighting that (1) adaptation is necessary, and (2) adaptation must occur at the appropriate abstraction level—the token space where modern LLMs operate.
>
> The comparison cleanly disentangles where utility loss comes from:
> - **SentinelLM-WhiteBox vs. SentinelLM**: Isolates the effect of access limitation
> - **AlienLM vs. ROT13**: Isolates the importance of learned vocabulary-level bijection vs. fixed character-level substitution
> - **AlienLM vs. vocab-sub**: Isolates the contribution of API-only adaptation
>
> We believe these expanded baselines provide a comprehensive evaluation within the deployable black-box API setting that AlienLM targets.

---

> ### Author Response · Authors · 2025-11-21
>
> ### A3. Existing Ablations and Improved Sign-posting
>
> We agree that component-wise attribution should be clearer. All key ablations are already included in Sec. 4 / Appendix; in the revision we will surface them more prominently and tighten cross-references to make each component's contribution immediately identifiable.
>
> ---
>
> ***A3.1 Encryption Ratio ρ Ablation*** We already provide a dedicated ablation in **Sec. 4.4 "Effect of Encryption Ratio on Performance"** with Figure 2 and full per-benchmark results in **Appendix Table 7**. This sweep over $\rho \in \{0.2, 0.3, 0.6, 0.8, 1.0\}$ shows a clear, monotonic privacy–utility trade-off:
>
> - Decreasing $\rho$ preserves more original tokens and improves utility
> - $\rho = 1$ provides maximum encryption coverage at the cost of utility
> - The trade-off is consistent across all benchmarks and models
>
> In the revision, we will explicitly reference Sec. 4.4 from Sec. 3 (Method) to make this ablation easier to locate when readers first encounter the ρ parameter.
>
> ---
>
> ***A3.2. API-only Fine-tuning (EAT) Ablation*** The **vocab-sub baseline** (bijection applied without any adaptation) is exactly an ablation that isolates the contribution of API-only fine-tuning (EAT):
>
> - **vocab-sub**: Applies vocabulary-level bijection to frozen original model → 38.25% relative performance
> - **AlienLM**: Applies vocabulary-level bijection + API-only adaptation (EAT) → 81.70% relative performance
>
> This 43.45 percentage point gap (38.25% → 81.70%) demonstrates that naïvely encrypting text collapses performance, and that most utility recovery comes from API-only adaptation. As shown in **Global Response Section G4.1**, this result is reported across all backbones and benchmarks.
>
> We will highlight this more explicitly as an ablation in the revised paper, clarifying that vocab-sub serves as the "no adaptation" control that isolates EAT's contribution.
>
> ---
>
> ***A3.3. Bijection Design / Construction Ablation*** We already isolate the contribution of principled bijection design in **Appendix A.5 (Ablation Study), Table 6**. This table compares target-based representations, proxy-based representations, and random permutation. Target-based representations use the target model's LM head for bijection construction, while proxy-based representations use a proxy model's embeddings (when target internals are unavailable). Random permutation is a naive vocabulary shuffling without optimization.
> - Target LM head achieves highest average accuracy (55.07%)
> - Proxy embeddings achieve competitive performance (53.32%, only -1.75 points), indicating that proxy representations capture sufficient similarity structure for effective bijection construction under black-box constraints
> - Random vocabulary permutation leads to severe degradation (36.96% average), confirming that naive substitution without bijection optimization fails to maintain utility
>
> In the revision, we will add a clear pointer to Appendix A.5/Table 6 from Sec. 3 and briefly summarize these trends in the main text, making it immediately clear that the bijection construction strategy is a key component that has been systematically ablated.
>
> ---
>
> ***A3.4. Summary*** Overall, the requested ablations are present in the current submission:
> - $\rho$ ablation**: Sec. 4.4, Figure 2, Appendix Table 7
> - **EAT ablation**: vocab-sub baseline (Global Response G4.1)
> - **Bijection design ablation**: Appendix A.5, Table 6
>
> The main issue is sign-posting. Our revision will surface these results more prominently and improve cross-referencing so readers can immediately identify each component's contribution without needing to search through the paper.
>
> ---
>
> We thank the reviewer again for the constructive feedback. In summary, we will (i) re-structure Sec. 3 for readability with intuition-first explanations, (ii) clarify baseline applicability under our client-only black-box threat model with expanded comparisons, and (iii) surface existing ablations more prominently through improved sign-posting. We believe these changes directly address the concerns while strengthening clarity and empirical grounding.

---

### Official Review · Reviewer_PNXp · 2025-11-06

**Soundness:** 2
**Presentation:** 3
**Contribution:** 2
**Rating:** 2
**Confidence:** 4

**Summary:**

The paper introduces AlienLM, a framework for privacy-preserving interaction with black-box LLM APIs by reinterpreting encryption as language translation. It constructs an “Alien Language” via a vocabulary-level bijection and adapts models through API-only fine-tuning (Encryption Adaptation Training, EAT). Experiments show AlienLM retains ~81% of original performance across seven benchmarks and four LLM backbones, outperforming naïve substitution and a simplified SentinelLM baseline. The authors claim robustness against token-mapping and frequency-analysis attacks and highlight tunable privacy–utility trade-offs.

**Strengths:**

- This paper works without white-box access, aligning with real-world constraints for commercial LLM APIs.
- Evaluates across multiple models, benchmarks, and includes ablations on encryption ratio and domain-specific fine-tuning.
- Greedy k-NN heuristic scales to large vocabularies and is computationally tractable.
- Clear articulation of privacy–utility trade-off.

**Weaknesses:**

- The “encryption” claim is misleading. The method is a deterministic substitution cipher with no formal cryptographic guarantees (e.g., IND-CPA), conflicting with NIST standards.

- Threat model and security evaluation are incomplete. The paper ignores known/chosen-plaintext attacks and parallel-corpus alignment, which break substitution systems.

- Strong alternatives like tokenizer-free models (CANINE[2], ByT5[3], Charformer[4]) are missing; SentinelLM [1] comparison is weakened.

**Questions:**

Given the strengths, I have the following concerns:

- While the method could be better framed as a prompt obfuscator for cloud LLMs, the claimed “encryption” provides no modern cryptographic security guarantees (e.g., IND‑CPA), which limits impact for actual compliance or regulated use-cases. Importantly, the method is a deterministic monoalphabetic substitution at the subword level. Classic results show such ciphers are inherently vulnerable to known‑plaintext and chosen‑plaintext/ciphertext attacks and provide no semantic security; the paper offers no formal analysis under standard definitions or an explicit adversary model. This significantly questions the "encryption" claim of the paper. How do you justify calling this “encryption” when it lacks formal properties like semantic security? Could you provide proofs?

- The evaluations omit realistic known‑plaintext scenarios (e.g., leakage of any plaintext–alientext pairs through logs, user reports, or outputs), which would allow rapid mapping recovery. Reported “robustness” focuses on nearest‑neighbor alignment in embedding/LM‑head space or corpus‑level frequency matching on mismatched proxies, not on parallel‑text alignment, n‑gram/LM‑based decipherment, or oracle‑based adaptive attacks. Can the authors include these experiments to test the "robustness" of the proposed method?

-  In its current form, there is no latency/throughput/serving cost analysis (client translation cost, server decoding/retokenization behavior, API tokenization pitfalls), no key‑rotation experiments, and no safety/alignment regressions assessment.

- If the goal is privacy via obfuscation under black‑box APIs, a natural baseline is to compare against tokenizer‑free/byte‑level LMs (e.g., CANINE, ByT5, Charformer) that are robust to noise, spelling changes, and non‑standard scripts—often with no fine‑tuning or with lightweight adaptation. These are omitted. Can the authors compare stronger baselines: (i) SentinelLM in a white‑box setting (clearly acknowledging it is out‑of‑scope for API‑only, but useful for calibrating the upper bound); (ii) CANINE/ByT5/Charformer as target models; (iii) alternative span‑level reversible transformations.



[1] Mishra et al. SentinelLMs: Encrypted Input Adaptation and Fine-Tuning of Language Models for Private and Secure Inference

[2] Clark et al. Canine: Pre-training an Efficient Tokenization-Free Encoder for Language Representation

[3] Xue et al. ByT5: Towards a Token-Free Future with Pre-trained Byte-to-Byte Models

[4] Tay et l. Charformer: Fast Character Transformers via Gradient-based Subword Tokenization

---

> ### Author Response · Authors · 2025-11-21
>
> We sincerely thank the reviewer for the thorough and constructive feedback. Below, we respond point-by-point to the four major questions raised in the review, highlighting concrete revisions and new experiments that we will incorporate into the final version.
> For context and consolidated results, we refer to our Global Response to Reviewers (shared across all reviewer replies), which provides:
>
> - G1: Conservative reframing and explicit threat model specification
> - G2: Expanded security/decipherment evaluation (LLM-based attacks, MT attacks, n-gram frequency analysis with known-plaintext leakage)
> - G3: Safety regression analysis across multiple models and benchmarks
> - G4: Comprehensive baseline comparisons (SentinelLM white-box upper bound, naive substitution baselines)
> - G5: Presentation and notation improvements
>
> In this response, we address each of your questions directly and reference relevant sections of the Global Response where detailed tables and analyses are provided.
>
>
> ---
>
> ### A1. Clarifying the Positioning: A Practical Privacy-Preserving Layer for API-based LLMs
>
> We acknowledge that the term "encryption" can be misleading when interpreted under modern cryptographic standards. In the revised paper, we will consistently position AlienLM as a **deployable, reversible translation-based privacy-preserving layer** for black-box LLM APIs, rather than as a cryptographic primitive claiming formal security guarantees (e.g., IND-CPA, semantic security).
>
> Concretely, AlienLM should be understood as a **deployable obfuscation mechanism** that (i) losslessly translates plaintext into an "Alien Language" to reduce human readability and casual leakage, and (ii) aligns the target model to this transformed distribution via API-only adaptation, thereby preserving utility under real API constraints.
>
> **Why existing secure computation methods are infeasible for commercial APIs:**
>
> We will add a brief comparison outlining why existing secure computation methods, while offering formal guarantees, are not feasible for commercial LLM APIs:
>
> - **Homomorphic encryption**: 1,000–10,000× computational overhead; requires white-box access to model architecture
> - **MPC/2PC protocols**: Prohibitive communication cost; incompatible with inaccessible model parameters
> - **TEEs (Trusted Execution Environments)**: Require specialized hardware not offered by major API providers (GPT, Claude, Gemini)
>
> In contrast, AlienLM provides **deployability, negligible overhead, and compatibility** with current API-only environments, which is the core application scenario we target. We will revise the paper to make this framing explicit and avoid conflating our approach with formal cryptographic primitives.
>
> ---
>
> ***A1.1. Explicit Threat Model Definition*** We will add a dedicated **Threat Model** subsection as described in **Global Response Section G1**. This subsection will explicitly specify the scope of in-scope adversaries (API-side eavesdropper/provider, realistic leakage scenarios, white-box surrogate attacks) and out-of-scope attacks (chosen-plaintext attacks requiring client-side compromise). Please refer to **G1** for the complete threat model specification.
>
> Our primary focus is on **API-side eavesdroppers/providers without client compromise**, which represents the most realistic threat for deployed black-box API scenarios.

---

> ### Author Response · Authors · 2025-11-21
>
> ### A2. Expanded Attack Evaluation
>
> We appreciate the reviewer's concern regarding the limited scope of our original security evaluation. To address this, we conducted a comprehensive set of decipherment experiments targeting scenarios where an adversary gains access to partial plaintext–alientext pairs or attempts to infer the mapping through large-scale LLM reasoning, MT-based attacks, and classical frequency analysis.
>
> ---
>
> ***A2.1. LLM-based Decipherment Attacks*** To evaluate AlienLM under powerful LLM-based attacks, we tested three state-of-the-art LLMs (GPT-5.1, GPT-5-mini, GPT-4.1) in both parallel and non-parallel corpus settings, approximating (i) leakage of small aligned plaintext–alientext snippets, and (ii) purely unaligned observation conditions.
>
> **Please see Global Response Tables G2-1 and G2-2** for complete results across all models and shot settings (0-shot, 1-shot, 5-shot, 20-shot). Even with 20-shot parallel examples, the recovered text remains far from intelligible:
> - BLEU stays at 1.88–11.56, ROUGE-L below 0.25
> - LLM-Judge scores remain around 1.0–1.7 out of 3.0, showing minimal semantic/structural overlap with original text
>
> These results indicate that bijection recovery does not trivially collapse under LLM reasoning alone at realistic leakage scales. We provide a representative failure case where GPT-5.1 asked to decode a math problem about diesel gallons instead produces unrelated content about probability histograms, demonstrating that attackers consistently generate fluent but semantically disconnected output.
>
> This represents empirical robustness within our specified threat model (API-side eavesdropper with limited leakage), not formal cryptographic security.
>
> ---
>
> ***A2.2. Multilingual Translation Model Attacks*** To further examine alignment-based vulnerabilities, we trained an NLLB-200-3.3B multilingual translation model as a direct sequence-to-sequence "decoder," emulating a realistic scenario where an adversary uses a strong translation model to map alientext to plaintext.
>
> **Please see Global Response Table G2-3** for detailed metrics. The specialized MT model also fails to recover meaningful plaintext (BLEU ≈ 11.4, ROUGE-L ≈ 0.29, LLM-Judge overall ≈ 1.0), with performance comparable to GPT-5.1 with 20-shot examples. This indicates that supervised seq2seq training provides no substantial advantage over in-context learning for this task.
>
> ---
>
> ***A2.3. N-gram Frequency Analysis (Proxy Corpora)*** To directly address classical frequency-based decipherment concerns, we evaluated AlienLM using n-gram statistics (n = 2, 3, 4) on three large proxy corpora (AceReason, Tulu3, SlimOrca) with optimal Hungarian bipartite matching over the top 1,000 tokens. This represents the computational upper bound for frequency analysis, with $O(V^3)$ complexity over $|V| \approx 128k$.
>
> **Please see Global Response Table G2-4** for complete results across all corpora and n-gram sizes. Across all proxy corpora and n-gram sizes, attack success remained extremely low:
> - Token-level accuracy stayed below 0.12% (fewer than 2 correct out of 1,000)
> - Bijection-level accuracy was effectively 0%, with only a single correct prediction out of 1,000 token pairs
> - Increasing n-gram size (2→4) or using larger corpora provided no measurable advantage
>
> These results suggest that frequency-based reconstruction with optimal bipartite matching yields negligible success rates at the scales tested, even when attackers have access to large proxy corpora. The large vocabulary space (128k tokens) may contribute to reducing the effectiveness of statistical signals that classical frequency analysis typically exploits in small-alphabet ciphers, though we acknowledge that substantially more sophisticated attack strategies remain unexplored.

---

> ### Author Response · Authors · 2025-11-21
>
> ***A2.4. N-gram Analysis with Parallel-Corpus Leakage (Known-Plaintext)*** We extended the analysis to realistic known-plaintext scenarios where attackers observe limited plaintext–alientext pairs (e.g., from API logs, user-shared debugging snippets). For each leakage level (10, 50, and 1,000 pairs), we exposed attackers to all alien tokens appearing in the parallel corpus and measured whether they could extrapolate to recover additional mappings among the top 1,000 most frequent tokens using the same Hungarian n-gram matching strategy.
>
> **Please see Global Response Table G2-5** for complete results.
> - Even with 1,000 parallel pairs exposing ~19,673 tokens, extrapolation to recover additional mappings remained negligible: token-level accuracy at 0.18–0.22%, bijection-level accuracy at 0%
> - Increasing leakage from 10 to 1,000 pairs (100× increase) provided no improvement in extrapolation success within the tested token range
>
> Overall, these results suggest that for practical leakage levels in real API deployments, recovering the full bijection is not straightforward. Note that our attacker already uses an optimal and computationally heavy recovery step, Hungarian bipartite matching with $O(V^3)$ complexity over a ~128k vocabulary, yet reconstruction still stays near zero in this regime.
>
> **Note** We acknowledge that any token appearing in leaked plaintext–alientext pairs is trivially revealed. Our experiments measure how well attackers can extrapolate beyond the leaked vocabulary to recover the full bijection.
>
> **Limitations** We leave stress-tests at substantially larger scales (e.g., million-pair leakage or stronger oracle-based settings) as future work and will state this limitation explicitly in the revision.
>
> ---
>
> ### A3. Practical Deployment Considerations
>
> We added deployment-focused analyses covering translator overhead, end-to-end serving latency/throughput, key rotation capabilities, and safety/alignment regressions after API-only adaptation. Detailed results are provided in **Global Response Section G3**.
>
>
> ---
>
> ***A3.1. Translator Overhead and End-to-End Serving Cost*** The only additional overhead introduced by AlienLM relative to standard API usage comes from the client-side translator used for alienization and recovery. This translator operates on CPU and performs simple dictionary lookups.
>
> **Please see Global Response Tables G3-1 and G3-2** for complete experiments results.
> - **Client-side overhead**: The translator consumes only 22.23 MB of CPU memory and introduces sub-millisecond latency per sample (0.21 ms for encryption, 0.35 ms for recovery), making it negligible compared to typical API network latency and model inference time.
> - **End-to-end performance**: Comparing original vs. AlienLM pipelines shows no meaningful difference in latency (62.13s vs. 61.82s), throughput (34.15 vs. 34.39 tokens/sec), or memory usage (15.32 GB for both).
> - **Practical implication**: AlienLM can be integrated into production API workflows without introducing measurable serving-time overhead, as the dominant costs remain network communication and remote model inference rather than local translation.
>
> ---
>
> ***A3.2. Key Rotation*** In our setting, key rotation corresponds to regenerating a new bijection. Section 4.6 in our original submission already evaluates this capability via seed-based bijection diversity with bucketed pairing: changing only the random seed yields highly distinct mappings with very low pairwise token overlap (maximum overlap $approx$ 1.96%), while utility remains stable across seeds (variance Var $approx$ 0.978 on the seven-benchmark average, with all seeds maintaining >78% relative performance on most benchmarks).
>
> This shows that AlienLM naturally supports practical key rotation/diversification—users can refresh bijections to obtain new Alien Languages without materially changing the utility profile.

---

> ### Author Response · Authors · 2025-11-21
>
> ***A3.3. Safety and Alignment Preservation***
>
> We evaluated whether AlienLM preserves safety and alignment behaviors after adaptation. Specifically, we tested four LLM backbones (LLaMA3-8B, Qwen2.5-7B, Qwen2.5-14B, Gemma2-9B) on a comprehensive suite of eight safety benchmarks covering harmfulness, toxicity, jailbreak resistance, and trustworthiness.
>
> **Please see Global Response Table G3-3** for complete safety/alignment benchmark results across all models.
>
> AlienLM is primarily designed to provide practical privacy preservation for black-box API usage, rather than to improve safety or alignment. The results indicate that applying AlienLM leads to safety degradation across multiple benchmarks, with average scores dropping from 45.67–70.49 to 39.30–42.32 across models.
>
> **Consistency with prior work** This behavior aligns with findings from Qi et al. (ICLR 2024), which systematically demonstrated that fine-tuning aligned LLMs—even on benign data for non-adversarial objectives—can unintentionally weaken safety guardrails. AlienLM's adaptation objectives prioritize utility preservation under obfuscated inputs, not safety retention.
>
> **Privacy–utility–safety trade-off** These results highlight a three-way tension in the design space. While AlienLM successfully preserves utility (~81% of original performance on downstream tasks) and provides empirical privacy benefits (see G2), safety characteristics degrade in the process. Since AlienLM is designed as a privacy-preserving layer for black-box API interactions rather than a safety enhancement mechanism, these regressions represent a known limitation that users should carefully consider when deploying the system in production environments.
>
> **Mitigation directions** We will explicitly note this privacy-safety interaction in the revised paper and discuss potential mitigation strategies as future work. Potential approaches include (1) safety-aware adaptation objectives that explicitly preserve refusal behaviors, (2) post-hoc safety re-alignment on alientext-transformed harmful prompts, or (3) hybrid client-side filtering before alienization.
>
> ---
>
> ### A4. Comprehensive Baseline Comparisons
>
> We expanded baselines in both white-box and black-box regimes. SentinelLM-WhiteBox is included as an explicit upper bound (see **Global Response Section G4**), and we add a tokenizer-free sanity-check baseline (ByT5 + Unicode ROT) to contextualize robustness outside the tokenized API ecosystem.
>
> ---
>
> ***A4.1. SentinelLM White-Box Upper Bound and Naive Substitution Baselines*** To contextualize AlienLM's performance, we compare against both upper-bound and lower-bound baselines on LLaMA3-8B across seven standard benchmarks.
>
> **Please see Global Response Section G4.1 and the corresponding table** for complete baseline comparisons including SentinelLM-WhiteBox, SentinelLM (black-box), vocab-sub, and ROT13.
>
> - **Black-box performance**: AlienLM achieves 81.70% of original utility, substantially outperforming SentinelLM (57.06%) in the same black-box setting.
>
> - **Gap to white-box upper bound**: AlienLM achieves approximately 84.82% of the SentinelLM-WhiteBox performance, demonstrating that API-only adaptation can preserve most utility despite lacking access to model parameters and gradients.
>
> - **Necessity of proper adaptation**: Both vocab-sub (38.25%) and ROT13 (36.03%) baselines collapse, highlighting that (1) adaptation is necessary, and (2) adaptation must occur at the appropriate abstraction level—the token space where modern LLMs operate—rather than at the character level without accounting for tokenizer boundaries and subword structure.

---

> ### Author Response · Authors · 2025-11-21
>
> ***A4.2. Character-Level Substitution Baseline with a Tokenizer-Free Model***
>
> To address the reviewer's suggestion regarding tokenizer-free and byte-level models, we include a tokenizer-free baseline as a sanity check to provide additional context. However, we emphasize that this comparison is **fundamentally outside the scope of our work**, which specifically targets privacy-preserving interaction with **currently deployed tokenizer-based commercial APIs** (e.g., GPT-4/5, Claude-3/4, Gemini).
>
> Tokenizer-free models (ByT5, CANINE, Charformer) differ substantially in architecture, training objectives, and—most critically—are not available as production black-box APIs that practitioners can access today. Our evaluation of ByT5 therefore serves as an architectural sanity check rather than a deployment-relevant baseline comparison.
>
> Specifically, we evaluate a character-level ROT transformation on ByT5-small, apply ROT at the Unicode level, and compare its behavior to AlienLM in an API-like setting. For fairness, we use the same LLaMA3-8B-based translator as in the main experiments when training and evaluating AlienLM.
>
> **Table:** Tokenizer-free/Byte-level Baseline with Unicode-level ROT Transformation
>
> | Method   | ARC-easy | ARC-challenge | HellaSwag | WinoGrande |
> | -------- | -------- | ------------- | --------- | ---------- |
> | Original | 29.00    | 16.98         | 27.83     | 46.49      |
> | ROT      | 28.70    | 18.34         | 27.85     | 48.62      |
> | AlienLM  | 26.52    | 20.73         | 27.59     | 50.67      |
>
>
> ByT5-Small with ROT shows small improvements on some tasks (e.g., WinoGrande +2.13 points), but the gains are modest and not consistent across benchmarks. AlienLM is competitive and in fact outperforms this tokenizer-free baseline on ARC-challenge (+2.39 points) and HellaSwag, despite operating through an external translation layer and a black-box target model.
> - **Performance comparison**: While character-level robustness helps ByT5 maintain performance under ROT transformation, the improvements are limited. AlienLM achieves comparable or better performance through vocabulary-level bijection and API-only adaptation, even when restricted to black-box access.
>
> - **Architectural considerations**: Tokenizer-free models operate at fundamentally different granularities (characters/bytes vs. subword tokens) and are trained with different objectives, making direct comparisons challenging. Our ByT5 experiment serves primarily as a sanity check to contextualize AlienLM's approach within the broader landscape of perturbation-robust architectures.
>
> - **Future exploration**: We plan to explore additional comparisons with other byte/character-level architectures (e.g., CANINE, Charformer) where feasible to further contextualize these results in the revision.
>
> We emphasize that AlienLM targets deployable privacy for prevailing tokenizer-based commercial APIs, while acknowledging tokenizer-free models as complementary directions for future research.
>
> ---
>
> We hope these clarifications and added experiments address the reviewer's concerns. We will incorporate the above reframing, threat-model statement, and new evaluations into the final revision.
>
> **References**
> - Qi, X., Zeng, Y., Xie, T., Chen, P.-Y., Jia, R., Mittal, P., Henderson, P. (2024). Fine-tuning Aligned Language Models Compromises Safety, Even When Users Do Not Intend To! Proceedings of ICLR 2024.

---

### Author Response · Authors · 2025-11-21
**Global Response to Reviewers: Revisions and Expanded Evaluation**

We thank the reviewers for their careful and constructive feedback. Below we provide a consolidated "Global Response" that summarizes the main revisions, clarifies the threat model and framing, and lists the new experiments and key insights that will be integrated into the final version. Individual reviewer replies reference this section where relevant.

---

### G1. Conservative Framing and Explicit Threat Model

Across reviews, a central concern was the security framing. We acknowledge that AlienLM should not be interpreted as modern cryptographic encryption under IND-CPA/semantic-security definitions. In the revision, we will consistently reframe AlienLM as a **deployable, reversible translation-based privacy-preserving layer for black-box LLM APIs**. Its guarantees are empirical and threat-model-bounded, not formal secrecy.

We will add a dedicated **Threat Model** subsection (end of Introduction / start of Sec. 3), explicitly specifying:

- **Primary in-scope adversary (API-side eavesdropper/provider)**: observes only alientext exchanged with the service (including outputs), with no access to client-side translator/seed key or model internals/weights.
- **Extended in-scope adversary (realistic leakage)**: obtains a small number of plaintext–alientext pairs (e.g., logs, user-shared snippets).
- **Upper-bound in-scope adversary (white-box surrogate)**: gains access to an AlienLM weight and attempts bijection recovery.
- **Out of scope**: chosen-plaintext attacks requiring compromise of the client-side translator or exposure of the bijection seed/key, which would trivially reveal the vocabulary mapping by design.

We will also explicitly state the deployment scope: AlienLM is intended for settings where remote API use is already permitted under policy/consent/compliance, and should not be viewed as bypassing categorical no-upload rules.

---

### G2. Expanded Security/Decipherment Evaluation

To address concerns about missing known-plaintext, parallel-corpus, and classical decipherment risks, we added four new attack suites beyond the original nearest-neighbor token-mapping attacks:

***G2.1 LLM-Based Inverse-Translation Attacks (Parallel / Non-Parallel)*** To evaluate AlienLM's robustness against powerful LLM-based decipherment, we tested three state-of-the-art language models (GPT-5.1, GPT-5-mini, GPT-4.1) as attackers attempting to reconstruct plaintext from alientext. These experiments directly address the concern that modern LLMs might leverage their vast linguistic knowledge and reasoning capabilities to infer the bijection mapping.

We evaluate two realistic leakage scenarios that correspond to plausible real-world API deployment risks:

- **Parallel-corpus leakage**: The attacker observes modest quantities of aligned plaintext–alientext pairs (e.g., from API logs, user-shared debugging snippets, or accidentally exposed examples). We test 0-shot (no examples), 1-shot, 5-shot, and 20-shot in-context learning to measure how much aligned evidence an LLM requires to perform effective inverse translation.

- **Non-parallel settings**: The attacker observes only unaligned plaintext and alientext corpora without direct correspondence between individual samples. This represents a weaker but still realistic adversarial condition where an attacker might collect alientext from network traffic and attempt to match it against publicly available plaintext corpora.

---

> ### Author Response · Authors · 2025-11-21
>
> We evaluate two realistic leakage scenarios that correspond to plausible real-world API deployment risks:
>
> - **Parallel-corpus leakage**: The attacker observes modest quantities of aligned plaintext–alientext pairs (e.g., from API logs, user-shared debugging snippets, or accidentally exposed examples). We test 0-shot (no examples), 1-shot, 5-shot, and 20-shot in-context learning to measure how much aligned evidence an LLM requires to perform effective inverse translation.
>
> - **Non-parallel settings**: The attacker observes only unaligned plaintext and alientext corpora without direct correspondence between individual samples. This represents a weaker but still realistic adversarial condition where an attacker might collect alientext from network traffic and attempt to match it against publicly available plaintext corpora.
>
> We evaluate using BLEU, ROUGE, and LLM-Judge (1–3 scale: 1 = no correspondence, 3 = perfect match, Judge Model: GPT-5.1) across semantic, structural, and overall dimensions.
>
> ```Table G2-1```: LLM inverse-translation attacks (Parallel Corpus)
>
> | Attack Model | N-Shots | BLEU | ROUGE-1 | ROUGE-2 | ROUGE-L | Semantic | Structural | Overall |
> |:------------|:-------:|:-----:|:--------:|:--------:|:--------:|:------------------------:|:--------------------------:|:-----------------------:|
> | GPT-5.1| 0| 3.22  | 0.26| 0.05 | 0.16 | 1.12| 1.07| 1.07|
> | GPT-5.1| 1| 6.16  | 0.33 | 0.08 | 0.19| 1.17| 1.15| 1.16|
> | GPT-5.1| 5| 9.50  | 0.38 | 0.12 | 0.23| 1.46| 1.38| 1.16|
> | GPT-5.1| 20| 11.56 | 0.40 | 0.15 | 0.25| 1.53| 1.59| 1.54|
> | GPT-5-mini| 0| 1.46 | 0.19 | 0.03 | 0.14 | 1.05| 1.03| 1.04|
> | GPT-5-mini| 1| 0.79| 0.17| 0.03| 0.11 | 1.06| 1.03| 1.04|
> | GPT-5-mini| 5| 2.17| 0.23 | 0.05| 0.15| 1.14| 1.13| 1.11|
> | GPT-5-mini| 20| 1.88| 0.23 | 0.05 | 0.15 | 1.17| 1.19| 1.17|
> | GPT-4.1  | 0 | 1.19  | 0.16 | 0.02 | 0.ƒ11 | 1.03| 1.00| 1.01|
> | GPT-4.1  | 1 | 2.63  | 0.23 | 0.04 | 0.14 | 1.07| 1.08| 1.06|
> | GPT-4.1  | 5 | 5.82  | 0.31 | 0.07 | 0.19| 1.18| 1.15| 1.11|
> | GPT-4.1  | 20  | 4.64  | 0.29 | 0.07 | 0.20| 1.13| 1.17| 1.11|
>
> ```Table G2-2```: LLM inverse-translation attacks (Non-Parallel Corpus)
>
> | Attack Model | N-Shots | BLEU | ROUGE-1 | ROUGE-2 | ROUGE-L | Semantic | Structural | Overall |
> |:------------|:-------:|-----:|--------:|--------:|--------:|------------------------:|--------------------------:|-----------------------:|
> | GPT-5.1  | 1 | 6.31  | 0.34 | 0.08 | 0.19 | 1.25 | 1.23  | 1.22  |
> | GPT-5.1  | 5 | 9.74  | 0.39 | 0.13 | 0.24 | 1.47 | 1.50  | 1.45  |
> | GPT-5.1  | 20  | 10.18 | 0.37 | 0.13 | 0.23 | 1.65 | 1.70  | 1.61  |
> | GPT-5-mini  | 1 | 0.86  | 0.16 | 0.02 | 0.11 | 1.03 | 1.03  | 1.01  |
> | GPT-5-mini  | 5 | 2.08  | 0.22 | 0.04 | 0.15 | 1.10 | 1.12  | 1.08  |
> | GPT-5-mini  | 20  | 1.86  | 0.24 | 0.06 | 0.16 | 1.14 | 1.18  | 1.13  |
> | GPT-4.1  | 1 | 1.82  | 0.22 | 0.03 | 0.13 | 1.06 | 1.05  | 1.03  |
> | GPT-4.1  | 5 | 4.00  | 0.29 | 0.07 | 0.18 | 1.21 | 1.21  | 1.15  |
> | GPT-4.1  | 20  | 4.13  | 0.28 | 0.07 | 0.19 | 1.12 | 1.14  | 1.08  |
>
>  Even with 20-shot parallel examples, reconstructions remain unintelligible across all metrics:
>
> - **Quantitative results**: BLEU stays at 1.88–11.56 (well below typical adequacy ranges reported in MT literature), ROUGE-L below 0.25, and LLM-Judge scores near minimum at 1.0–1.7 out of 3.0.
>
> - **Qualitative failure mode**: In a representative case, GPT-5.1 asked to decode a math problem about diesel gallons instead produces unrelated content about probability histograms (BLEU 1.72, ROUGE-L 0.15, exact match 0, all LLM-Judge scores = 1.0):
>   ```
>   - Original plaintext: "How many gallons are in a ton of diesel if the density of diesel is 37.7 kilograms per gallon? A ton is 2000 pounds. … 2000 / 83.23 = 24.03 gallons. The answer is 24."
>
>   - Alientext: "You }. all Lis unin || Walton goofy Blitz assertThat … AllanTranslator Eis hire324/lo"
>
>   - GPT-5.1 output (20-shot): "You are given two histograms, one for Walton and one for Elgg, showing the distribution of scores on the game Blitz. … the required probability is 0.43"
>   ```
>
>   Attackers consistently generate fluent but semantically disconnected output, hallucinating plausible content based on surface-level token associations without recovering underlying meaning.
> - **Bijectin Recovery**: These results indicate that bijection recovery does not trivially collapse under LLM reasoning alone at realistic leakage scales. The bijection successfully disrupts semantic neighborhoods that LLMs rely on for language understanding.
>
> - **Empirical Robustness**: This represents empirical robustness within our specified threat model (API-side eavesdropper with limited leakage), not formal cryptographic security. Substantially larger leakage volumes or more sophisticated attacks remain untested.

---

> ### Author Response · Authors · 2025-11-21
>
> ***G2.2 Multilingual MT Decoder Attacks*** We train a strong NLLB-200-3.3B translator as a direct seq2seq attacker.
>
> ```Table G2-3```: MT-based inverse-translation attack
>
> | BLEU | ROUGE-1 | ROUGE-2 | ROUGE-L | Semantic | Structural | Overall |
> |:-----:|:--------:|:--------:|:--------:|:------------------------:|:--------------------------:|:-----------------------:|
> | 11.40 | 0.32  | 0.18  | 0.29  | 1.04 | 1.07  | 1.00  |
>
> The specialized MT model also fails to recover meaningful plaintext despite dedicated training. Performance is comparable to GPT-5.1 with 20-shot examples (BLEU 11.40 vs. 11.56), indicating that supervised seq2seq training provides no substantial advantage over in-context learning for this task.
>
> ---
>
> ***G2.3 Classical N-gram Frequency Analysis (Proxy Corpora)*** We evaluated classical frequency-based decipherment using n-gram statistics (n = 2, 3, 4) on three large proxy corpora (AceReason, Tulu3, SlimOrca) with optimal Hungarian bipartite matching over the top 1,000 tokens. This represents the computational upper bound for frequency analysis, with $O(V^3)$ complexity over $|V| \approx$ 128k vocabulary.
>
> ```Table G2-4```: N-gram frequency analysis with proxy corpora
>
> | Reference Corpus | N-gram | Token level acc | Bijection level acc |
> | :----------------: | :------: | :---------------: | :-------------------: |
> | acereason  | 2  | 0.04%  | 0.00%  |
> | acereason  | 3  | 0.05%  | 0.01%  |
> | acereason  | 4  | 0.04%  | 0.00%  |
> | tulu3  | 2  | 0.06%  | 0.00%  |
> | tulu3  | 3  | 0.06%  | 0.00%  |
> | tulu3  | 4  | 0.05%  | 0.00%  |
> | slimorca  | 2  | 0.10%  | 0.00%  |
> | slimorca  | 3  | 0.12%  | 0.01%  |
> | slimorca  | 4  | 0.10%  | 0.00%  |
>
>
> - **Token-level accuracy** stays below 0.12% (fewer than 2 correct out of 1,000), and **bijection-level accuracy** is effectively 0% across all corpora and n-gram sizes at tested scales.
> - Increasing n-gram size (2→4) or using larger corpora shows minimal improvement in recovery rates.
> - The large vocabulary space (128k tokens) appears to weaken the statistical signals that classical frequency analysis typically exploits in small-alphabet ciphers, yielding negligible recovery rates under optimal matching at the tested scales.
>
> ---
>
> ***G2.4 N-gram Analysis with Parallel-Corpus Leakage (Known Plaintext)*** We extended the analysis to realistic known-plaintext scenarios where attackers observe limited plaintext–alientext pairs (e.g., from API logs or user-shared snippets). For each leakage level (10, 50, 1,000 pairs), we expose all alien tokens appearing in the parallel corpus to the attacker as known plaintext. Then we measure whether they can extrapolate to recover additional mappings among the top 1,000 most frequent tokens using the same Hungarian n-gram matching strategy.
>
> ```Table G2-5```: N-gram analysis with parallel-corpus leakage
>
> | \# parallel pairs | \# Known tokens | N-gram | Token level acc | Bijection level acc |
> | :-----------------: | :-------------------------: | :------: | :---------------: | :-------------------: |
> | 10 | 972 | 2  | 0.21%  | 0.00%  |
> | 10 | 972 | 3  | 0.22%  | 0.00%  |
> | 10 | 972 | 4  | 0.21%  | 0.00%  |
> | 50 | 3223  | 2  | 0.19%  | 0.00%  |
> | 50 | 3223  | 3  | 0.19%  | 0.00%  |
> | 50 | 3223  | 4  | 0.18%  | 0.00%  |
> | 1000  | 19673  | 2  | 0.19%  | 0.00%  |
> | 1000  | 19673  | 3  | 0.19%  | 0.00%  |
> | 1000  | 19673  | 4  | 0.18%  | 0.00%  |
>
> - Even with 1,000 parallel pairs exposing ~19,673 tokens, extrapolation to recover additional mappings among the top 1,000 frequent tokens remains negligible: token-level accuracy stays at 0.18–0.22%, bijection-level accuracy at 0%.
> - Increasing leakage from 10 to 1,000 pairs (100× increase) provides no improvement in extrapolation success within the tested token range.

---

> ### Author Response · Authors · 2025-11-21
>
> ### G3. Practical Deployment Considerations
>
> ***G3.1 Translator Overhead and End-to-End Serving Cost*** To assess real-world feasibility, we measured the computational overhead introduced by AlienLM's client-side translator (responsible for alienization and recovery) and compared end-to-end serving performance against the original API pipeline. The translator operates on CPU and performs simple dictionary lookups using huggingface tokenizers, while the target LLM remains on the API provider's infrastructure.
>
> ```Table G3-1```: Throughput/latency Original vs. AlienLM
>
> |  | original | alienlm |
> | -------------------- | :--------: | :-------: |
> | Latency (s) | 62.13 | 61.82  |
> | avg token per second | 34.15 | 34.39  |
> | memory usage (GB) | 15.32 | 15.32  |
>
> ```Table G3-2```: Translator-side overhead breakdown
> | Stage | Unit | Time (ms) |
> |:--|:--|:--:|
> | Alienization | per sample | 0.21 |
> | Alienization | per token | 0.00 |
> | Recovery | per sample | 0.35 |
> | Recovery | per token | 0.01 |
>
> | Resource | Value |
> |:--|--:|
> | Translator memory (MB) | 22.23 |
>
>
>
> - **Client-side overhead**: The translator consumes only 22.23 MB of CPU memory and introduces sub-millisecond latency per sample (0.21 ms for encryption, 0.35 ms for recovery), making it negligible compared to typical API network latency and model inference time.
>
> - **End-to-end performance**: Comparing original vs. AlienLM pipelines shows no meaningful difference in latency (62.13s vs. 61.82s), throughput (34.15 vs. 34.39 tokens/sec), or memory usage (15.32 GB for both).
>
> - **Practical implication**: AlienLM can be integrated into production API workflows without introducing measurable serving-time overhead, as the dominant costs remain network communication and remote model inference rather than local translation.
>
> ---
>
> ***G3.2 Safety/Alignment Preservation*** To examine whether AlienLM's API-only adaptation preserves safety and alignment behaviors, we evaluated four target models (LLaMA3-8B, Qwen2.5-7B, Qwen2.5-14B, Gemma2-9B) on a comprehensive suite of eight safety benchmarks covering harmfulness, toxicity, jailbreak resistance, and trustworthiness. All metrics are normalized to a 0–100 scale where higher values indicate safer behavior (100 = optimal safety).
>
> ```Table G3-3```: Safety/alignment results Original vs. AlienLM
>
> | Target Model | Method | WildG (↑) | HarmBench (↑) | ToxiGen (↑) | XSTest (↑) | WildJ-b (↑) | WildJ-h (↑) | DAN (↑) | TrustLLM (↑) | AVG (↑) |
> |:------------|--------:|--------:|--------:|--------:|--------:|--------:|--------:|--------:|--------:|--------:|
> | LLaMA3-8B | Original | 5.21  | 84.69  | 100.00  | 96.89  | 8.40  | 80.50  | 98.70 | 89.50  | 70.49 |
> | LLaMA3-8B | AlienLM  | 28.84 | 55.00  | 99.29  | 75.33  | 0.80  | 7.35  | 20.67 | 51.25  | 42.32 |
> | Qwen2.5-7B  | Original | 15.09 | 83.44  | 100.00  | 92.22  | 0.40  | 13.85  | 64.33 | 68.25  | 54.70 |
> | Qwen2.5-7B  | AlienLM  | 31.11 | 43.44  | 95.21  | 72.67  | 0.80  | 2.55  | 29.00 | 50.25  | 40.63 |
> | Qwen2.5-14B  | Original | 9.21  | 93.44  | 100.00  | 94.00  | 0.80  | 25.10  | 23.30 | 19.50  | 45.67 |
> | Qwen2.5-14B  | AlienLM  | 34.45 | 59.06  | 100.00  | 76.44  | 0.40  | 1.75  | 11.00 | 48.25  | 41.42 |
> | Gemma2-9B | Original | 10.41 | 92.81  | 100.00  | 90.00  | 1.60  | 42.35  | 32.33 | 80.25  | 56.22 |
> | Gemma2-9B | AlienLM  | 34.98 | 44.06  | 99.71  | 75.11  | 1.60  | 3.30  | 8.67  | 47.00  | 39.30 |
>
> ```Note. WildG = WildGuardTest; HarmBench = HarmBench; ToxiGen = ToxiGen ; XSTest = XSTest; TrustLLM = JailbreakTrigger (TrustLLM); DAN = Do-anything-now; WildJ-b / WildJ-h = WildJailbreak benign / harmful contrast sets.```

---

> ### Author Response · Authors · 2025-11-21
>
> **Key outcome**:
>
> - **Observed regressions**: AlienLM exhibits safety degradation across multiple benchmarks, with average scores dropping from 45.67–70.49 to 39.30–42.32 across models. The magnitude varies by metric: HarmBench drops 29.75–48.75 points, DAN resistance drops 23.66–90.03 points, while ToxiGen remains relatively stable (95.21–100.00).
>
> - **Consistency with prior work**: This behavior aligns with findings from Qi et al. (ICLR 2024), which systematically demonstrated that fine-tuning aligned LLMs—even on benign data for non-adversarial objectives—can unintentionally weaken safety guardrails.AlienLM's adaptation objectives prioritize utility preservation under obfuscated inputs, not safety retention.
>
> - **Privacy–utility–safety trade-off**: These results highlight a three-way tension in the design space. While AlienLM successfully preserves utility (~81% of original performance on downstream tasks, see G4.1) and provides empirical privacy benefits (see G2), safety characteristics degrade in the process. Since AlienLM is designed as a privacy-preserving layer for black-box API interactions rather than a safety enhancement mechanism, these regressions represent a known limitation that users should carefully consider when deploying the system in production environments.
>
> - **Mitigation directions**: We will note this privacy-safety interaction explicitly in the revised paper and investigate mitigation strategies as future work. Potential approaches include (1) safety-aware adaptation objectives that explicitly preserve refusal behaviors, (2) post-hoc safety re-alignment on alientext-transformed harmful prompts, or (3) hybrid client-side filtering before alienization.
>
> ---
>
> ### G4. Baselines and Novelty Clarification
>
> ***G4.1 Strong Baselines (White-Box Upper Bound + Naive Substitution)*** To contextualize AlienLM's performance, we compare against both upper-bound and lower-bound baselines on LLaMA3-8B across seven standard benchmarks. We include:
>
> - **SentinelLM-WhiteBox**: An idealized upper bound assuming full white-box access to model internals (gradients, weights, architecture), which is unavailable in the black-box API setting AlienLM targets.
> - **SentinelLM (black-box)**: The original black-box version for direct comparison which is introduced in the original submission.
> - **vocab-sub**: Naive vocabulary-level substitution without any model adaptation, applying the bijection directly to the frozen original model.
> - **ROT13**: Unicode-level character substitution without adaptation, isolating how much performance can be preserved when inputs are simply perturbed, representing an approach that does not account for the tokenizer-based nature of modern LLMs. but it si trained!
>
> | Method  | MMLU | ARC-e | ARC-c | HellaS | WinoGrande | TruthfulQA | GSM8k | Average | Ratio  |
> | ------------------- | ----- | ----- | ----- | ----- | ----- | ----- | ----- | ----- | ----- |
> | Original  | 67.32 | 84.13 | 59.39 | 57.07 | 74.35 | 35.25 | 75.89 | 64.77 | |
> | SentinelLM-WhiteBox | 63.67 | 80.77 | 53.50 | 54.16 | 72.61 | 37.58 | 74.45 | 62.39 | 96.33 |
> | SentinelLM | 29.92 | 46.34 | 27.56 | 38.47 | 55.09 | 30.23 | 31.08 | 36.96 | 57.06 |
> | vocab-sub  | 25.18 | 26.39 | 20.56 | 26.66 | 47.59 | 25.83 | 1.21  | 24.77 | 38.25 |
> | ROT13  | 22.92 | 25.00 | 20.05 | 25.36 | 49.09 | 20.93 | 0.00  | 23.34 | 36.03 |
> | AlienLM | 46.56 | 72.14 | 44.28 | 47.86 | 61.48 | 35.01 | 63.08 | 52.92 | 81.70 |
>
> - **Black-box performance**: AlienLM achieves 81.70% of original utility, substantially outperforming SentinelLM (57.06%) in the same black-box setting. This demonstrates that AlienLM's bijection construction and API-only adaptation strategy effectively preserves utility under realistic deployment constraints.
>
> - **Gap to white-box upper bound**: AlienLM recovers approximately 73.8% of the gap between naive vocab-sub (38.25%) and the white-box upper bound (96.33%). This shows that API-only adaptation can achieve substantial utility recovery despite lacking access to model internals.
>
> - **Naïve substitution vs. character-level failure**: vocab-sub (38.25%) collapses because it applies vocabulary-level substitution to a frozen model, disrupting semantic neighborhoods without adaptation. ROT13 (36.03%), despite being adapted, performs similarly poorly because it operates at the character level without accounting for tokenizer boundaries and subword structure that tokenizer-based LLMs rely on. These results highlight that (1) adaptation is necessary, but (2) adaptation must occur at the appropriate abstraction level—the token space where modern LLMs actually operate.

---

> ### Author Response · Authors · 2025-11-21
>
> ***G4.2 Novelty Relative to Prior Obfuscation*** To clarify AlienLM's contribution relative to prior obfuscation work, we acknowledge that the high-level goal of balancing opacity and utility has precedents in prior research (e.g., prompt obfuscation, adversarial perturbations, privacy-preserving text transformation). However, AlienLM's novelty lies in making this approach universally deployable under black-box API constraints at modern vocabulary scales:
>
> - **Single reusable vocabulary-level bijection**: Unlike per-instance rewriting methods that require generating new transformations for each input, AlienLM constructs a single bijection over $~10^5$ tokens that can be reused across all inputs (Pape et al., 2025; Lin et al., 2025). This enables practical key management, consistent translation, and efficient client-side implementation.
>
> - **API-only end-to-end alignment**: AlienLM achieves utility preservation through API-only adaptation (query/response interactions only), without requiring access to model weights, gradients, or internal representations (Mishra et al., 2024, Lin et al., 2024). This is critical for deployment on commercial APIs where white-box access is unavailable.
>
> - **Modern vocabulary scale**: Prior work on substitution-based obfuscation typically operates over small alphabets or limited vocabularies. AlienLM demonstrates that bijection-based obfuscation can be effectively applied and aligned at the scale of modern tokenizers (~128k tokens), a regime where classical frequency analysis becomes computationally impractical (see G2.3–G2.4) (Pape et al., 2025; Lin et al., 2025; Lin et al., 2024).
>
> We will sharpen this distinction in the Related Work section and Method overview of the revised paper, explicitly positioning AlienLM's contributions within the broader landscape of privacy-preserving text transformation techniques.
>
> ---
> ### G5. Presentation/Notation Fixes
>
> We will improve clarity and consistency by:
> - explicitly separating token set vs. index set and clarifying the role of $I_\rho$
> - standardizing notation (e.g., using $\tau_\text{tgt}$ consistently for the target model subscript),
> - adding design-intent paragraphs and goal–objective correspondences in Sec. 3,
> - correcting typos noted by reviewers and running a full proofreading pass.
>
> ---
>
> We hope this consolidated summary clarifies our revisions and the empirical basis for our claims. We will integrate these changes, tables, and insights into the final paper, and keep individual reviewer responses focused and self-contained while aligned with this global framing.
>
> **References**
> - Carlini, Nicholas, et al. “Is Private Learning Possible with Instance Encoding?” 2021 IEEE Symposium on Security and Privacy (SP), IEEE, 2021, pp. 410–427. DOI: 10.1109/SP40001.2021.00099.
> - Qi, Xiangyu, et al. “Fine-tuning Aligned Language Models Compromises Safety, Even When Users Do Not Intend To!” Proceedings of the International Conference on Learning Representations (ICLR), 2024. arXiv:2310.03693.
> - Mishra, Abhijit, Mingda Li, and Soham Deo. “SentinelLMs: Encrypted Input Adaptation and Fine-tuning of Language Models for Private and Secure Inference.” Proceedings of the AAAI Conference on Artificial Intelligence, vol. 38, no. 19, AAAI Press, 2024, pp. 21092–21100. DOI: 10.1609/aaai.v38i19.30136.
> - Pape, David, et al. “Prompt Obfuscation for Large Language Models.” arXiv, 2024, arXiv:2409.11026.
> - Lin, Sam, et al. “EmojiPrompt: Generative Prompt Obfuscation for Privacy-Preserving Communication with Cloud-based LLMs.” Proceedings of the 2025 Conference of the Nations of the Americas Chapter of the Association for Computational Linguistics: Human Language Technologies (NAACL-HLT 2025), vol. 1 (Long Papers), Association for Computational Linguistics, 2025, pp. 12342–12361.
> - Lin, Yalan, et al. “CodeCipher: Learning to Obfuscate Source Code Against LLMs.” arXiv, 2024, arXiv:2410.05797.

---

### Note · Program_Chairs · 2026-01-17
**Submission Desk Rejected by Program Chairs**

The following references in this submission do not refer to real documents and/or have major errors in bibliographic information:

 X. Yao et al. Federated learning for large language models: A survey. arXiv, 2024. URL https: //arxiv.org/abs/2402.01639. Survey.
A. Mishra et al. Sentinellms: Training language models to process encrypted or obfuscated inputs. arXiv, 2024a. URL https://arxiv.org/. Preprint; to-verify venue/authors.
J. Remy et al. Trans-tokenization: Tokenizer transfer and alignment across llms. arXiv, 2024. URL https://arxiv.org/. Preprint; to-verify.
Z. Lin et al. Codecipher: Obfuscation-aware code models via token substitution and embedding perturbation. arXiv, 2024. URL https://arxiv.org/. Preprint; to-verify title/author list.